# Q-NeRF: Neural Radiance Fields on a Simulated Gate-based Quantum Computer

## Abstract

Recently, Quantum Visual Fields (QVFs) have shown promising improvements in model compactness and convergence speed for learning 2D images. Meanwhile, novel-view synthesis has seen major advances with Neural Radiance Fields (NeRFs), where models learn a compact representation from 2D images to render 3D scenes, albeit at the cost of large models and intensive training. In this work, we extend the approach of QVFs by introducing *QNeRF*, the first hybrid quantum-classical model designed for novel-view synthesis from 2D images. QNeRF leverages parameterized quantum circuits to encode spatial and view-dependent information via quantum superposition and entanglement, resulting in more compact models. We present two architectural variants. *Full QNeRF* maximally exploits all quantum amplitudes to enhance representational capabilities. In contrast, *Dual-Branch QNeRF* introduces a task-informed inductive bias by branching spatial and view-dependent quantum state preparations, drastically reducing the complexity of this operation and ensuring scalability and potential hardware compatibility. Our experiments demonstrate that—when trained on images of reduced resolution—QNeRF matches or outperforms classical NeRF baselines while using less than half the number of parameters. These results suggest that Quantum Machine Learning can serve as a competitive alternative for continuous signal representation in high-level tasks in Computer Vision, such as 3D representation learning.

## 1 Introduction

Neural Radiance Fields (NeRFs) have revolutionised novel view synthesis by modelling 3D scenes as a continuous volumetric function implicitly parametrized—in their simplest form—by a multi-layer perceptron (MLP) (Mildenhall et al., 2020; Barron et al., 2021; Gao et al., 2022). By encoding scene geometry and appearance through radiance and density fields, NeRFs achieve photorealistic rendering of unseen views in complex environments with compact representations. These methods have found applications in areas such as 3D scene reconstruction (Wang et al., 2021; Noguchi et al., 2021; Chen et al., 2021), human modelling (Park et al., 2021; Peng et al., 2023; Karunratanakul et al., 2020), and image processing (Kasten et al., 2021; Li et al., 2021). Despite their impressive generalisation across novel viewpoints, the performance of NeRF-based frameworks remains heavily dependent on the underlying learning model, particularly in terms of the training efficiency and convergence speed. While increasing the capacity of classical neural networks remains a standard strategy for improving performance, a more principled alternative lies in rethinking the foundations of learning models.

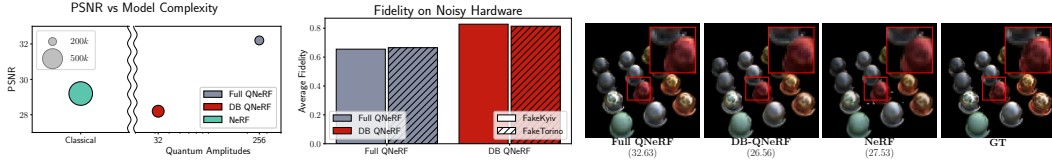

|        (a) PSNR vs. complexity        |        (b) Average Fidelity        |  (c) Novel-view synthesis on the *materials* scene  |

Figure 1: 1a: comparison between model complexity as number of parameters (dot size) and number of amplitudes encoded with PSNR. 1b: average fidelity on noisy simulated hardware for the proposed models. 1c: example reconstruction with PSNR of the highlighted box. Zoom recommended.

In this work, we investigate Quantum Neural Networks (QNNs) as a novel, previously unexplored and highly efficient framework in the context of novel-view synthesis. Quantum Machine Learning (QML) has recently emerged as an alternative paradigm to classical Machine Learning with theoretical promises and hopes to enhance characteristics of the ML models (Biamonte et al., 2017; Cerezo et al., 2022). Following early theoretical results demonstrating advantages in expressivity and learnability (Abbas et al., 2021; Liu et al., 2021; Pirnay et al., 2024), the field is increasingly shifting towards practical implementations. In particular, QNNs showed evidence of quantum advantage with respect to both model compactness and training efficiency in specific learning settings (Simoes et al., 2023; Cherrat et al., 2024; Landman et al., 2022). Recently, the introduction of Quantum Visual Fields (QVFs) (Wang et al., 2025) showed the practical advantage of a QML model on the task of learning 2D field representations. Following these advancements, we introduce a **Quantum Neural Radiance Field** (**QNeRF**), a hybrid classical-quantum model in which the standard MLP used in NeRF-based approaches is replaced with a more expressive QNN. By investigating the effects of QML components on volumetric rendering from 2D images, we aim to establish a foundational comparison between the original NeRF model and its quantum-enabled counterpart.

Inspired by QVFs, QNeRF uses a learnable classical embedding to efficiently represent local coordinates as a quantum state. Then, the state is processed by a sequence of trainable quantum operations (i.e., the QNN). Finally, classical information is extracted by the final state to reconstruct the novel view by standard volumetric rendering techniques. This design places the QNN at the model's core building block so the framework can leverage quantum resources (superposition, entanglement) to explore compact and expressive internal representations while remaining compatible with existing rendering pipelines and near-term quantum hardware.

We propose two architectural variants: (i) A **Full QNeRF** (Fig. 2a), which maximally exploits quantum resources to enhance representational capacity, and (ii) A **Dual-Branch QNeRF** (Fig. 2b), which encodes spatial and view-dependent coordinates as separate intermediate quantum states, leading to improved scalability, higher noise tolerance, and potentially better compatibility with near-term quantum hardware.

We experimentally demonstrate on noiseless simulated quantum hardware that, for low-resolution scenes, the proposed Full QNeRF model achieves higher reconstruction quality than the classical baseline while using less than half the number of parameters (see Fig. 1a and Fig. 1c). In contrast, the Dual-Branch model attains performance comparable to the classical baseline but exhibits consistently higher noise tolerance when evaluated on simulated noisy devices (Fig. 1b). These results indicate that quantum neural architectures can act as competitive building blocks for continuous signal representation in volumetric novel-view synthesis. In summary, our work has the following technical contributions:

- We introduce **QNeRF**, the first NeRF-based architecture designed with quantum hardware compatibility (Sec. 4). In simulated experiments on two different datasets, QNeRF achieves comparable or better performance than the classical NeRF baseline while using fewer than half the parameters.
- We propose a **dual-branch quantum embedding** that encodes spatial and view-dependent features as separate quantum states, providing a task-informed inductive bias and simplifying quantum state preparation (Sec. 4.1). In addition, we incorporate an output scaling mechanism that mitigates exponential concentration effects.

To encourage further research on this topic, we release the full implementation of our model[1] along with all experimental code and training details.

## 2 Related Work

### 2.1 Classical view synthesis

Neural networks have long been applied to image-based view synthesis (Flynn et al., 2019; Liu et al., 2019; Mildenhall et al., 2019). However, a major breakthrough in the field was the introduction of

---

[1]The source code will be released after the first round of review. A working demo can be found in the supplement.

Neural Radiance Fields (NeRF) (Mildenhall et al., 2020), which proposed learning a continuous volumetric scene representation from posed images. NeRF defines a 5D mapping from spatial location and viewing direction to colour and density, enabling high-fidelity novel view synthesis. The success of NeRF set a new standard for rendering quality and inspired a substantial body of subsequent research (Xie et al., 2022; Gao et al., 2022; Yariv et al., 2023; Li et al., 2023; Mildenhall et al., 2022a). Numerous extensions have aimed to improve rendering performance, generalisation, and robustness under specific conditions, such as Mip-NeRF (Barron et al., 2021), which improved performance at multiple resolutions, and instant-NGP (Müller et al., 2022), developed to reduce training time. It is worth noting that many of these NeRF variants retain the core architectural element of MLP to learn the radiance field, and could therefore benefit from a hybrid classical-quantum architecture such as the one we propose in this work. However, since our proposed quantum architecture enhances the "building block" of NeRF, these variants are only partially related to this work.

## 2.2 QUANTUM-ENHANCED COMPUTER VISION

Quantum Computer Vision (QCV) has recently emerged as a subfield of quantum machine learning, focusing on leveraging quantum computing for tasks in image processing, graphics, and 3D reconstruction. Early work in this domain predominantly relied on quantum annealers Seelbach Benkner et al. (2021); Heidari et al. (2024); Choong et al. (2023); Birdal et al. (2021); Farina et al. (2023); Zaech et al. (2022). More recently, attention has shifted toward gate-based quantum computing due to its greater generality and potential for quantum advantage. Gate-based approaches offer increased flexibility in designing parameterized quantum circuits, enabling variational models that are trainable via classical optimisation. Several QCV applications have been proposed using gate-based architectures, including quantum convolutional neural networks (Henderson et al., 2020; Fan et al., 2024; Hai et al., 2025), quantum generative adversarial networks (Huang et al., 2021; Silver et al., 2023), quantum autoencoders (Rathi et al., 2023), and other hybrid architectures (Gharibyan et al., 2025; Cherrat et al., 2024; Landman et al., 2022).

Of particular relevance to this work are two prior contributions. The first is (Zhao et al., 2024), which applies a hybrid architecture to visual data using a "sandwich" structure, where quantum layers are embedded between classical feature extractors and regressors. While effective, this configuration makes it difficult to disentangle the contribution of the quantum component from the overall model performance. In contrast, our architecture ensures that the quantum circuit processes the full encoded input and that the quantum embedding plays a direct role in model expressivity, allowing a cleaner analysis of the quantum contribution.

The second is the Quantum Visual Field (QVF) model Wang et al. (2025), which this work is inspired by. In this work, the authors proposed a hybrid architecture for visual data using amplitude embedding and variational quantum circuits. The proposed model was able to both lower parameter requirements compared to classical baselines, and also enable more efficient learning of high-frequency structures. While QVFs demonstrated the viability of using quantum embeddings for visual field representation, our work addresses the different and more challenging task of novel-view synthesis from 2D images, which requires several key innovations to work effectively: (i) our proposed model is explicitly tailored for novel-view synthesis by jointly processing positional and view-dependent features; (ii) we introduce a dual-branch encoding mechanism in Dual-Branch QNeRF, which reduces the complexity of amplitude embeddings and lowers the gate complexity of quantum state preparation (Zhang et al., 2022), while also introducing a task-aligned inductive bias that reflects the natural factorization of positional and directional information in view synthesis; (iii) we incorporate an output-scaling strategy (for both Full QNeRF and Dual-Branch QNeRF) to mitigate the *exponential concentration* phenomenon (Thanasilp et al., 2024; McClean et al., 2018; Larocca et al., 2025).

## 3 QUANTUM MACHINE LEARNING

In this section, we provide the necessary background on Quantum Machine Learning (QML). We defer notation details to App. B. Comprehensive introductions to Quantum Computing and QML can be found in (Nielsen & Chuang, 2010) and (Schuld et al., 2014), respectively.

Quantum Machine Learning arises from the intersection of Quantum Computing with Machine Learning and Deep Learning. The field encompasses a broad variety of models, methodologies, and

objectives (Peral-García et al., 2024; Mishra et al., 2021). One of the central goals of QML is to design models that exploit quantum phenomena such as entanglement and superposition to achieve an "advantage" in terms of computational resources, representational power, or learning performance when compared to purely classical methods (Cerezo et al., 2022). A prominent class of models within QML is that of Quantum Neural Networks (QNNs). In (Abbas et al., 2021), it was proved that QNNs can achieve higher effective dimensions than comparable classical neural networks, which translates into faster convergence during training. Following this theoretical result, experimental results from both general (Simoes et al., 2023) and vision-specific tasks (Cherrat et al., 2024; Landman et al., 2022) show that QNNs can achieve the same level or better performances than classical Neural Networks with fewer parameters and with increased convergence speed. Given these properties, it is natural to investigate the application of QNNs to high-dimensional vision tasks, where efficient training and generalization are particularly critical due to the high dimensionality and complexity of the models required.

**Quantum Neural Networks** In Quantum Machine Learning, the term Quantum Neural Network is often used interchangeably with Parameterized Quantum Circuit (PQC), as the two notions are closely related (Wan et al., 2016). A QNN typically consists of a sequence of quantum gates whose operations depend on the classical inputs $\mathbf{x}$ and on a set of free parameters $\theta$, which are optimized during the training process. Given an initial state $|\phi\rangle$ (e.g. the state $|0\rangle^{\otimes N}$, where $N$ is the size of the quantum system), the application of a parameterized quantum circuit $P$ results in the state $P(\theta; \mathbf{x})|\phi\rangle$. In many architectures, the circuit $P$ can be naturally decomposed into two stages: a data encoding stage and a variational (trainable) stage. The encoding is typically achieved through a fixed set of gates that map the classical input into a quantum state (often referred to as the *embedding* or *feature map*), denoted by $S(\mathbf{x})$. The variational stage, represented by $V(\theta)$, then acts on the embedded state. Thus, the final state of the circuit can be described as $V(\theta)S(\mathbf{x})|\phi\rangle = V(\theta)|\phi_S(\mathbf{x})\rangle$, where $|\phi_S(\mathbf{x})\rangle$ denotes the quantum state obtained by embedding the input $\mathbf{x}$. The structure $V(\theta)$ typically constitutes a variational ansatz, designed to be expressive enough to represent the target function during learning.

**Noisy Intermediate-Scale Quantum** Current quantum hardware operates in what is referred to as the *Noisy Intermediate-Scale Quantum* (NISQ) era (Preskill, 2018). In this regime, quantum devices are composed of tens to hundreds of qubits, but they are subject to non-negligible levels of noise and decoherence, which significantly limit their computational capabilities. As a consequence, it is of fundamental importance to design quantum circuits that minimize both the number of quantum gates and the overall circuit depth. Excessive gate count or depth can result in an accumulation of errors that rapidly degrade the fidelity of the computation, making the outcomes unreliable. This limitation motivates the development of shallow, hardware-efficient quantum circuits and encourages a careful trade-off between expressivity and robustness to noise.

## 4 QUANTUM NEURAL RADIANCE FIELDS (Q-NeRFs)

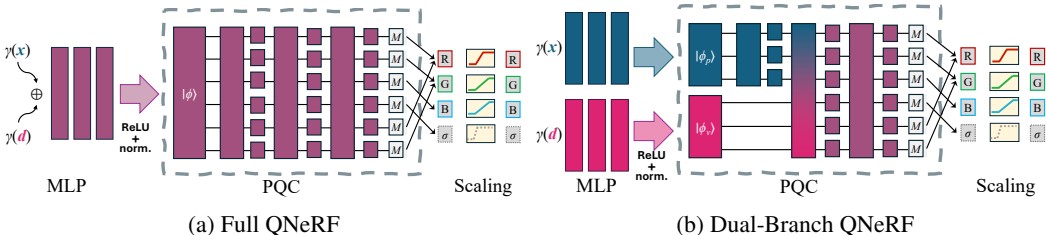

(a) Full QNeRF          (b) Dual-Branch QNeRF

Figure 2: Scheme of the proposed models for $n = 6$ qubits, and $\ell = 2$ repetitions. Positional and view-dependent coordinates (shown in teal and magenta, respectively) are first encoded into quantum amplitudes by one (a) or two (b) MLPs, processed by a PQC (dashed box). Each gate is coloured according to the information that is processed (purple, if the information depends on both positional and view-dependent features, or blue or magenta if it depends only on positional or view-dependent features). The multi-qubit purple gates in (a) and (b) represent a dense entangling layer (see Fig. 3, top), while the multi-coloured one in (b) is a partial entangling layer (see Fig. 3, bottom). Then, the state is converted to classical information through a parity-based measurement, and finally processed with a scaling layer to reconstruct the output view.

We describe our proposed method **Quantum Neural Radiance Field** (QNeRF), consisting of a hybrid quantum-classical model designed to efficiently represent scenes and generate novel views. A brief background on Neural Radiance Fields is provided in App. A. We start by describing how to encode classical coordinates into a quantum state. This description is given in two variants: one for the *Full* encoding, which allows to leverage all the properties of the quantum system, and one for the *Dual-Branch* encoding, which reduces the expressivity of the model to drastically increase scalability and compatibility with current quantum devices. A scheme of the proposed circuits is provided in Fig. 2. Then, we present the quantum circuit design choices. Finally, we describe postprocessing operations, consisting of a parity measurement to extract information, and then a "de-concentration" scaling layer that enhances the performance of the model by mitigating exponential concentration (Thanasilp et al., 2024).

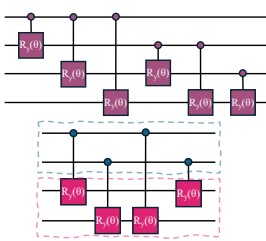

Figure 3: Dense entangling layer (top) and a partial entangling layer (bottom), for $n = 4$.

## 4.1 MLP-BASED QUANTUM EMBEDDING

Neural Radiance Field (Mildenhall et al., 2020) models heavily rely on positional encodings for both spatial and view-dependent information. This technique enriches each input coordinate $x \in \mathbb{R}$ by mapping it to a higher-dimensional space $\mathbb{R}^{2L}$ using the transformation

$$\gamma(x) = \left(\sin(2^0\pi x), \cos(2^0\pi x), \ldots, \sin(2^{L-1}\pi x), \cos(2^{L-1}\pi x)\right). \tag{1}$$

This encoding has been shown to be crucial for mitigating the spectral bias of neural networks, which tend to underrepresent high-frequency components, thereby enabling the representation of fine-grained variations in the data. Even for modest values of $L$, the dimension of $\gamma(x)$ exceeds the number of qubits available in current quantum devices. Therefore, to obtain a quantum state that represents the structure from the encoded vector, we employ amplitude embedding. Unlike angle embedding or unitary amplitude encoding (Johri et al., 2021), which scale linearly with the number of features, amplitude embedding offers exponential compression, as a vector with $2^n$ components is represented using only $n$ qubits.

**Full Embedding with All Amplitudes**   We first describe the *Full* embedding strategy in which all the possible amplitudes are used to encode data, obtaining a more expressive representation. As a first step, the enriched input vector $\gamma(\mathbf{x})$, obtained via positional encoding, is mapped to a normalised vector $\mathcal{M}(\mathbf{x})$ of $2^n$ *amplitudes*. This requires addressing three key challenges: (i) decoupling the dimensionality of $\mathbf{x}$ from the hardware-constrained number of amplitudes $2^n$; (ii) ensuring that $\mathcal{M}(\mathbf{x})$ retains sufficient representational structure for the downstream quantum model; and (iii) satisfying the normalization constraint $\sum_{i \leq 2^n} |\alpha_i|^2 = 1$, where $\alpha_i$ denotes the $i$-th amplitude. To this end, we employ a lightweight Multi-Layer Perceptron (MLP) with an output layer of size $2^n$. A ReLU activation function is applied at the end of the MLP to ensure that all amplitudes are non-negative and increasing sparsity, thereby facilitating quantum state preparation (Nakaji et al., 2022). Finally, the amplitudes are normalised to guarantee compatibility with quantum embedding requirements.

**Dual-Branch Embedding**   Although recent work has improved the efficiency of amplitude state preparation (Pagni et al., 2025), encoding $2^n$ amplitudes on a quantum device remains a significant challenge (Long & Sun, 2001; Plesch & Brukner, 2011; Zhang et al., 2022). Since the state preparation requires up to $\mathcal{O}(k)$ gates, where $k$ is the number of amplitudes, exploiting the full quantum space can be unfeasible on current NISQ devices (Preskill, 2018). On the other hand, there is no intrinsic reason to treat positional and view-dependent features identically within the quantum embedding. Drawing inspiration from the original NeRF architecture, in which view-dependent features are encoded after positional features have already been processed, we introduce a **dual-branch** that separates and independently processes these components, while exponentially reducing the number of amplitudes required. Formally, we divide the total qubit budget into $n_p$ and $n_v$ qubits, assigned to positional and view-dependent encodings, respectively. Two MLPs are trained to produce $2^{n_p}$- and $2^{n_v}$-dimensional vectors, which are independently amplitude-encoded and then composed into a tensor product state:

$$|\phi(x)\rangle = |\phi_p(x_p)\rangle \otimes |\phi_v(x_v)\rangle. \tag{2}$$

In the simplest case where $n_p = n_v = n/2$, the total number of amplitudes is reduced to $2^{n/2+1}$, offering an exponential reduction in $n$ (e.g., $2^n$ vs. $2^{n/2+1}$) relative to the Full QNeRF approach. Furthermore, the dual-branch strategy also yields an exponential reduction in the number of parameters required for the MLPs. In particular, our Dual-Branch QNeRF model can scale up to more than 15 qubits while maintaining fewer parameters than the classical NeRF model with approx. $590k$ parameters (see Fig. 1a). Amplitudes, parameters, and gates for different numbers of qubits are provided in Table 1.

| Qubits | Full QNeRF | | | Dual-Branch QNeRF | | |
|---|---|---|---|---|---|---|
| | Amplitudes | Parameters ($k$) | Gates | Amplitudes | Parameters ($k$) | Gates |
| 4 | 16 | 160 | 10 | 8 | 290 | 21 |
| 6 | 64 | 172 | 21 | 16 | 293 | 42 |
| **8** | **256** | **222** | **36** | **32** | **297** | **70** |
| 10 | 1 024 | 420 | 55 | 64 | 305 | 105 |
| 12 | 4 096 | 1 213 | 78 | 128 | 322 | 147 |

Table 1: Comparison of Full and Dual-Branch models in terms of amplitudes and parameters count as a function of the number of qubits. The number of parameters depends on the number of qubits, and is computed for an MLP of 3 layers with hidden dimension $h = 256$. The gate counts depend on the number of repetitions $\ell$, and are reported for $\ell = 1$ and $\ell = 2$ for Full and Dual-Branch, respectively. The number of qubits $n = 8$ (in bold) corresponds to the value used in Sec. 5. Note that the the the number of required amplitudes in the Full model scale quadratically compared to the DB model.

## 4.2 QUANTUM CIRCUIT DESIGN

Following quantum embedding, the encoded data is processed by a variational quantum circuit with learnable parameters $V(\boldsymbol{\theta})$. In line with prior work on real-valued quantum feature spaces (Wang et al., 2025), we restrict the variational ansatz to the real subspace of the Hilbert space by employing only $R_Y$ rotations. This design choice simplifies gradient-based optimisation and increases resilience to hardware noise, without significantly limiting model expressivity for the task considered. The circuit is composed of three elementary modules: (i) a single-qubit rotational layer consisting of $R_Y(\theta)$ gates applied independently to each qubit; (ii) a *dense entangling layer* that applies controlled-$R_Y(\theta)_{(i,j)}$ gates across each pairs of qubits $i > j$ (Fig. 3, top); (iii) a *partial entangling layer* that applies, for each $i$ in $n_p$ and for each $j$ in $n_v$, a controlled-$R_Y(\theta)_{(i,j)}$ (Fig. 3, bottom). In the Full QNeRF, we build a circuit as a sequence of $\ell \geq 1$ blocks, each composed of a dense entangling layer followed by a rotational layer over all $n$ qubits. A graphical representation for $\ell = 3$ on 6 qubits is given in Fig. 2a. In the Dual-Branch QNeRF, we first apply a full entangling layer between the $n_p$ qubits allocated for the positional features, followed by a rotational layer over the corresponding qubits. After these steps, the state of the system can be written as

$$(V_p(\theta) \otimes I_v)\left(|\phi_p(x_p)\rangle \otimes |\phi_v(x_v)\rangle\right) = V_p(\theta)|\phi_p(x_p)\rangle \otimes |\phi_v(x_v)\rangle, \tag{3}$$

where $V_p$ is the operator corresponding to the two positional layers. This design ensures that positional information is first internally processed and entangled, mirroring the sequential encoding strategy used in classical NeRFs. Next, the view-dependent amplitude embedding is introduced, and a partial dense entangling layer is applied between the $n_p$ positional qubits and the $n_v$ view-dependent qubits, combining the two feature spaces. A global rotational layer is then applied to all $n$ qubits. By entangling states corresponding to positional and view-dependent features, we ensure that the model can learn the correlation between all the input coordinates. Finally, to further enhance model expressivity, the circuit can optionally be extended with $\ell - 2$ additional blocks, each comprising a dense entangling layer followed by a rotational layer over all $n$ qubits. A scheme of the model for $\ell = 3$ on 6 qubits is given in Fig. 2b.

## 4.3 PARITY-BASED MEASUREMENTS

At the end of a quantum circuit, the quantum state is typically converted into classical data via measurement. In this work, we measure the system in the computational basis, corresponding to projective measurements in the eigenbasis of the Pauli-$Z$ operator. To mitigate the effects of barren

plateaus (McClean et al., 2018; Larocca et al., 2025), we employ local measurements. Specifically, we define a family of single-qubit observables acting on individual qubits as

$$\hat{O}_i = \mathbb{I}^{\otimes(i-1)} \otimes Z \otimes \mathbb{I}^{\otimes(n-i)}, \quad \text{for } i = 1, \ldots, n, \tag{4}$$

where $Z$ denotes the Pauli-$Z$ operator and $\mathbb{I}$ is the identity operator on a single qubit. In practice, we write $\hat{O}_i = Z_i$, indicating a Pauli-$Z$ measurement on the $i$-th qubit. These local projective measurements yield bitstrings corresponding to independent measurements on each qubit. Local observables are both hardware-efficient and less susceptible to barren plateaus compared to global observables, and have been shown to enhance the trainability of variational quantum algorithms (Cerezo et al., 2021; Thanasilp et al., 2023). The output of the circuit is given by the expectation value $\mathcal{O}(\mathbf{x}) = \langle \phi_{\text{in}} | V(\boldsymbol{\theta})^\dagger \hat{O} V(\boldsymbol{\theta}) | \phi_{\text{in}} \rangle$, where $\mathbf{x}$ is the input (derived from positional and view-dependent features), $V(\boldsymbol{\theta})$ is the parameterized quantum circuit, and the input state $|\phi_{\text{in}}\rangle = |\phi_p\rangle \otimes |\phi_v\rangle$ is prepared via a dual-branch amplitude embedding. The output $\mathcal{O}(\mathbf{x})$ is therefore a vector of $n$ real numbers in the interval $[-1, 1]$. To obtain outputs suitable for the considered task, we transform $\mathcal{O}(\mathbf{x})$ into a 4-dimensional vector representing RGB colour channels and a volumetric density. This transformation proceeds in two stages. First, for each of the four output components, we select a subset of qubits $\mathcal{C}_i \subset \{1, \ldots, n\}$ and compute the average of their corresponding expectation values, i.e., $\tilde{o}_i(\mathbf{x}) = \frac{1}{|\mathcal{C}_i|} \sum_{j \in \mathcal{C}_i} \mathcal{O}_j(\mathbf{x})$, where $\mathcal{O}_j(\mathbf{x})$ is the expectation value associated with qubit $j$. This step, referred to as *parity averaging*, aggregates information from specific qubit subsets into semantically meaningful outputs. Finally, we clip the resulting values to $[0, 1]$ The resulting vector $\mathbf{o}(\mathbf{x}) = (o_1(\mathbf{x}), \ldots, o_4(\mathbf{x})) \in [0, 1]^4$ can then be used to reconstruct a novel view as described in (Mildenhall et al., 2020). The novel view is then used to compute the standard MSE loss by comparing it with the ground truth.

## 4.4 Output Scaling

Variational quantum circuits are known to suffer from the *exponential concentration* phenomenon (Thanasilp et al., 2024), which refers to the tendency of their output distributions to concentrate exponentially around their mean as the number of qubits increases. This effect significantly limits the expressive power of such circuits, particularly in representing distributions with high variance. To mitigate the limitations introduced by exponential concentration and improve the trainability of our model, we introduce

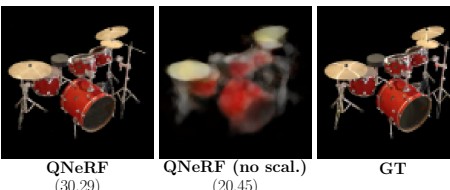

**QNeRF**
(30.29)
**QNeRF (no scal.)**
(20.45)
**GT**

Figure 4: Visualisation of the effect of output scaling for a Full QNeRF, after 50 epochs (PSNR reported in brackets).

a learnable scaling factor $\alpha_c$ applied to the output associated with each channel. The final output $\alpha_c \mathcal{O}_c(\mathbf{x})$ is then clipped to $[0, 1]$. This modification widens the range of output values, thereby counteracting the tendency of the circuit outputs to collapse around their mean, and acts in practice as a "de-concentration" layer. Empirically, we found that output scaling plays a crucial role in enhancing model performance, in particular with regard to the predicted density (see Fig. 4).

## 5 Experimental Evaluation

To evaluate our proposed approach, we conduct experiments on noiseless, simulated quantum hardware across multiple scenes, repeating each experiment on 5 different seeds. (Sec. 5). These experiments are performed with the *Pennylane* framework (Bergholm et al., 2018). In addition, we evaluate the effect of noise on the proposed architecture by computing fidelity between ideal outputs, and outputs obtained after transpilation on realistic hardware and noisy gate implementations and measurements (Sec. 5.1). These experiments were carried out with IBM's *qiskit* framework. To assess the effectiveness of QNeRF in rendering novel views, we select a representative subset of eight scenes: four from the Blender dataset, obtained from a synthetic setting, and four from the LLFF dataset (Mildenhall et al., 2019), following the evaluation protocol established in (Mildenhall et al., 2020). All images used during training and testing are downscaled to reduce training time, as quantum simulation is computationally expensive, and currently, there are no GPU-based simulation frameworks compatible with our proposed method. In particular, training a QNeRF model required up to 50 hours of CPU time. Additional training details are provided in App.D.

Finally, it is important to emphasise that, as this represents the first QML-based approach for novel-view synthesis, direct comparison with other baselines is inherently challenging. To establish a meaningful, foundational comparison, we compare both Full and Dual-Branch QNeRF with the original classical NeRF architecture. A summary of the results is provided in Tables 2a and 2b, which summarise peak signal-to-noise ratio (PSNR). We provide details and visualizations in App. F.

The encoding sizes for positional and view-dependent features are set to 10 and 4, respectively, following the original NeRF model. In both quantum models, the encoding MLPs consist of three fully connected layers with a hidden dimension of 256, and output equal to the number of amplitudes, corresponding to $2^8$ for the Full QNeRF, and $2^5$ for the Dual-Branch QNeRF. Both ansätze are constructed as described in Sec. 4.2 with $\ell = 1$ and $\ell = 2$ for Full and Dual-Branch models, respectively. For both models, we selected the smallest meaningful value of $\ell$ (Dual-Branch model requires at least 2 layers, to include interaction between positional and view-dependent features). Initial values for the quantum parameters are chosen using *identity initialisation* (Grant et al., 2019), a common initialisation strategy to mitigate barren plateaus. To balance model expressivity with computational complexity, we selected models with 8 qubits. Finally, the parity measurement assigns 2 qubits for each output channel.

**Blender Dataset** For the Blender dataset, we selected the scenes *materials*, *ficus*, *lego*, and *drums*. Each image was downscaled via average pooling to a resolution of $100\times100$ pixels. All models were trained on a fixed subset of 100 training images and evaluated on a fixed validation set of 200 samples. Complete quantitative results are reported in Table 2a. On this dataset, we observe that the Full QNeRF model can outperform the classical baseline on each scene, with an average of 2 dB more (31.59 vs 29.53). Dual-Branch has a PSNR slightly lower than the classical one ($\approx 0.7$ below).

| Model | *Materials* | *Ficus* | *Lego* | *Drums* | Average |
|---|---|---|---|---|---|
| Full QNeRF | **33.88 ± 0.16** | **30.26 ± 0.21** | **34.47 ± 0.04** | **28.07 ± 0.05** | **31.67 ± 0.11** |
| DB QNeRF | 29.94 ± 0.31 | 28.59 ± 0.27 | 31.32 ± 0.25 | 25.63 ± 0.19 | 28.87 ± 0.26 |
| Class. Baseline | 29.90 ± 0.19 | 29.74 ± 0.17 | 31.79 ± 0.16 | 26.70 ± 0.13 | 29.53 ± 0.16 |

(a) Blender

| Model | *Trex* | *Room* | *Horns* | *Fern* | Average |
|---|---|---|---|---|---|
| Full QNeRF | **22.87 ± 1.04** | **27.94 ± 0.45** | **23.45 ± 0.71** | **23.21 ± 0.41** | **24.37 ± 0.65** |
| DB QNeRF | 22.03 ± 0.42 | 26.12 ± 0.51 | 22.02 ± 0.48 | 22.22 ± 0.42 | 23.10 ± 0.46 |
| Class. Baseline | 22.11 ± 0.40 | 26.68 ± 0.50 | 21.02 ± 0.58 | 21.96 ± 0.37 | 22.94 ± 0.46 |

(b) LLFF

Table 2: Final PSNR (dB) of the proposed models compared to the classical baseline. Each value is averaged over 5 seeds and presented with the standard deviation. To prevent overfitting, the training is stopped as soon as the testing PSNR starts decreasing, or after 50 epochs.

**LLFF dataset** For the LLFF dataset, we selected the scenes *horns*, *fern*, *trex*, and *room*. Each image was downscaled to a resolution of $63 \times 47$. For each run, the set of images was split into training and testing sets in an 80-20 split. Complete quantitative results are reported in Table 2b. The results on this dataset are similar to the ones on the Blender dataset, with minor differences. First, the average PSNR is in general lower, as the represented scenes are more complex. As a consequence, the gap between the Full QNeRF and the NeRF performances is lower (slightly more than 1 dB). As before, the Dual-Branch model has a similar performance to the classical model, but this time it performs slightly better. The standard deviation on each experiment is higher: this depends on the fact that each seed has different splits, as in contrast to the Blender dataset, there is no fixed split.

### 5.1 NOISE RESILIENCE

We now analyse the noise resilience of the proposed models under hardware-realistic assumptions, for different numbers of ansatz repetitions. Specifically, we employ the noise models *FakeKyiv* and *FakeTorino* provided by IBM (IBM Quantum, 2025), which approximate the effects observed on real quantum devices. For more details and additional results, we refer to the App. G. To quantify

resilience to hardware noise, we compute the state fidelity between the noiseless circuit output and the corresponding noisy output obtained after: (i) amplitude state preparation; (ii) transpilation into the native gate set of the hardware; (iii) implementation of noisy quantum gates; (iv) noisy measurements. The results are reported in Fig. 5. The highlighted values, corresponding to $\ell = 0$, represent the fidelity of the standard amplitude embedding implemented in Qiskit (Javadi-Abhari et al., 2024). As reported in Sec. 4, the Dual-Branch encoding requires exponentially fewer amplitudes, and exhibits therefore substantially lower error rates for corresponding values of $\ell$. The results for the configuration used in this work ($\ell = 1$ for Full, $\ell = 2$ for Dual-Branch) are also depicted in Fig. 1b.

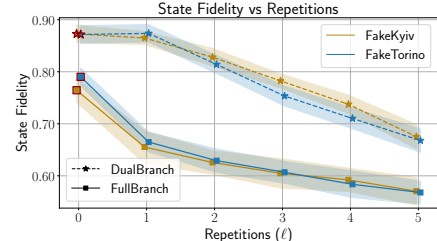

From the plot, we observe that the Dual-Branch model consistently achieves higher fidelity than the Full Branch model. It is important to note that the fidelities reported here are severely under-optimized, as no advanced compilation or noise mitigation strategies were applied. In practical executions on real hardware, several techniques are typically employed, including optimized transpilation strategies (Murali et al., 2019; Li et al., 2019), live error mitigation methods such as Pauli Twirling (Wallman & Emerson, 2016; Tsubouchi et al., 2025), and post-processing approaches such as HAMMER (Tannu et al., 2022). Moreover, the model may, in principle, partially adapt to systematic errors during training. Therefore, the state fidelities shown

Figure 5: State fidelity for random parameter initializations, evaluated over 50 runs for different 8-qubit ansatz and noise models. The red marks (corresponding to $\ell = 0$) indicate the fidelity of the amplitude state preparation.

here represent a conservative, worst-case estimate of noise resilience. Even modest optimizations in state preparation, such as approximate amplitude embedding (Nakaji et al., 2022), can already yield significant improvements in fidelity.

## 6 DISCUSSION AND LIMITATIONS

The experimental results indicate that the Full QNeRF model consistently outperforms the NeRF baseline with less than half of the parameters. The Dual-Branch model has a reduced performance (comparable with the classical model), again with a reduced parameter count, but presents a much higher noise tolerance (Fig. 5) due to the branched encoding and the partial entangling layer. One limitation of our current study is the computational cost required to perform large-scale quantum simulations. Currently, training QNeRF models requires substantially more time than classical baselines due to the overhead of simulating quantum circuits on a CPU. We expect this gap to shrink as both quantum simulators and hardware accelerators improve, and with the integration with GPU-accelerated quantum simulators (e.g. (Stein et al., 2024; Schieffer et al., 2025)). On the other hand, deploying these models on real-world quantum hardware may introduce challenges depending on sampling and hardware noise, limited hardware connectivity, and the complexity of gradient computation, which must be addressed. It is important to note that this work focuses on the fundamental challenge of introducing the gate-based QML paradigm into volumetric neural novel-view systems. While it has many limitations at this exploratory stage, we believe it opens up a new research direction with many potential improvements and advances in future.

## 7 CONCLUSION

This work introduced two hybrid quantum-classical models for novel-view synthesis: Full QNeRF and Dual-Branch QNeRF. While QVFs had previously demonstrated the potential of quantum embeddings for 2D field representations, here we confirmed and extended these properties in a new and more challenging setting. Our experiments show that Full QNeRF consistently outperforms the classical NeRF baseline, resulting in $+7\%$ and $+6\%$ PSNR over Blender and LLFF datasets. On the other hand, Dual-Branch QNeRF shows similar performances to the classical baseline ($-2\%$ and $+1\%$ respectively), but with a more scalable and noise-resistant architecture (more than $0.8$ state fidelity without any kind of error mitigation technique), showing promising compatibility with current, or near-future quantum hardware. While many challenges remain, we believe these results represent an early but concrete step toward assessing the potential of quantum-enhanced representations for

volumetric novel-view synthesis, marking a step forward toward practical quantum advantage in vision-based learning.

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

APPENDIX

We provide a brief introduction on Neural Radiance Fields (Appendix A) and on Quantum Computing (Appendix B). In Appendix C we discuss the scalability (in terms of amplitudes to encode and number of parameters) of the proposed quantum models. In Appendix D we provide some details of implementation and training. We conduct an ablation study in Appendix E, in which we evaluate a "de-quantised" QNeRF model, showing that the QNN is a fundamental component of the proposed model. We discuss relevant metrics for novel-view rendering in Appendix F, also providing some results and visualisations on the Blender dataset. In Appendix G, we discuss the experimental details for the noise evaluation discussed in Appendix 5.1, and we provide additional details and statistics on the circuits transpiled on *FakeKyiv* and *FakeTorino*. In Appendix H we provide an ablation study on the number of qubits. Then, in Appendix I we compute the expected execution time on *FakeTorino* hardware. In Appendix J and Appendix K we evaluate respectively trainability (by estimating the gradient), and noise resilience during inference. Finally, in Appendix L and Appendix M we evaluate how QNeRF can be integrated with other tasks and models, respectively: in particular, in L we perform the task of Mesh Extraction for the models trained on novel-views rendering, and in M we evaluate the integration of QNeRF with Mip-NeRF (Barron et al., 2021).

## A BACKGROUND ON NERF

This section provides a brief background on the idea behind NeRF. For an in-depth introduction, we refer to (Xie et al., 2022).

### A.1 SCENE REPRESENTATION

Neural Radiance Fields model a 3D scene as a continuous volumetric function that maps spatial locations and viewing directions to emitted radiance and volume density. Formally, a scene is represented by a function

$$F : (\mathbf{x}, \mathbf{d}) \mapsto (\mathbf{c}, \sigma), \tag{5}$$

where $\mathbf{x} = (x, y, z) \in \mathbb{R}^3$ denotes the spatial coordinate, $\mathbf{d} = (\theta, \phi)$ represents the viewing direction, $\mathbf{c} = (r, g, b) \in [0, 1]^3$ denotes the RGB colour at that point along the viewing ray, and $\sigma \in \mathbb{R}_{\geq 0}$ represents the volume density, corresponding to the opacity at that location.

In practice, this function is approximated using a multilayer perceptron, or more generally, a neural network, trained to fit a sparse set of 2D images of a scene by minimising a photometric reconstruction loss.

### A.2 VOLUME RENDERING

The colour of a pixel in an image is synthesised by integrating the radiance accumulated along a camera ray as it traverses the 3D scene. Given a ray parameterised as $\mathbf{r}(t) = \mathbf{o} + t\mathbf{d}$, where $\mathbf{o}$ is the ray origin (i.e., the camera centre) and $\mathbf{d}$ is the viewing direction (corresponding to a unit vector in $\mathbb{R}^3$), the colour $C(\mathbf{r})$ of the ray is computed using the volume rendering equation:

$$C(\mathbf{r}) = \int_{t_n}^{t_f} T(t)\, \sigma(\mathbf{r}(t))\, \mathbf{c}(\mathbf{r}(t), \mathbf{d})\, dt, \tag{6}$$

where $\sigma(\mathbf{r}(t))$ denotes the volume density at point $\mathbf{r}(t)$, $\mathbf{c}(\mathbf{r}(t), \mathbf{d})$ is the emitted RGB colour in direction $\mathbf{d}$, and $[t_n, t_f]$ are the near and far bounds of the ray.

The term $T(t)$ represents the accumulated transmittance, that is, the probability that a photon travels from $t_n$ to $t$ without being absorbed, and is defined as:

$$T(t) = \exp\left(-\int_{t_n}^{t} \sigma(\mathbf{r}(s))\, ds\right).$$

In practice, the continuous integral is approximated numerically using a discrete sampling scheme. The ray is divided into $N$ stratified or uniform intervals $\{t_i\}_{i=1}^{N}$, and the colour is estimated using quadrature:

$$\hat{C}(\mathbf{r}) = \sum_{i=1}^{N} T_i \left(1 - \exp(-\sigma_i \delta_i)\right) \mathbf{c}_i,$$

where $\sigma_i = \sigma(\mathbf{r}(t_i))$, $\mathbf{c}_i = \mathbf{c}(\mathbf{r}(t_i), \mathbf{d})$, $\delta_i = t_{i+1} - t_i$ is the distance between adjacent samples, and $T_i = \exp\left(-\sum_{j=1}^{i-1} \sigma_j \delta_j\right)$ denotes the discrete approximation of the transmittance.

Finally, this rendering formulation enables gradient-based optimisation of the scene representation by comparing synthesised views with ground-truth images, making it possible to reconstruct detailed volumetric radiance fields from sparse observations.

## B BACKGROUND ON QUANTUM COMPUTING

This section presents some key concepts of Quantum Computing. For a detailed introduction, we refer to (Nielsen & Chuang, 2010).

A *qubit* is a two-level quantum system described by a complex-valued unit vector in a 2-dimensional Hilbert space:

$$|\psi\rangle = \alpha|0\rangle + \beta|1\rangle, \quad \text{where } \alpha, \beta \in \mathbb{C}, \quad |\alpha|^2 + |\beta|^2 = 1,$$

and $|0\rangle, |1\rangle$ represent the quantum analogue of classical bit values 0 and 1.

Similarly, an $n$-qubit system can be represented by a $2^n$-dimensional Hilbert space:

$$|\psi\rangle = \sum_{i=0}^{2^n-1} \alpha_i|i\rangle, \quad \text{where } \sum_i |\alpha_i|^2 = 1.$$

Given two quantum states $|\phi\rangle$ and $|\psi\rangle$ of respectively $n$ and $m$ qubits, the state $|\phi\rangle \otimes |\psi\rangle$, where $\otimes$ represents the tensor product, can be represented in a quantum system of size $n + m$.

### B.1 QUANTUM GATES AND CIRCUITS

Quantum computations are realised via the application of unitary transformations to quantum states. A unitary operator $U \in \mathbb{C}^{2^n \times 2^n}$ evolves a quantum state $|\psi\rangle$ to a new state $|\psi'\rangle = U|\psi\rangle$. Physically, such operations are implemented as *quantum circuits*, which can be decomposed into a sequence of elementary operations known as *quantum gates*.

In practice, most hardware platforms support only one- and two-qubit gates. Nevertheless, these gates form a universal set, meaning they can be composed to approximate any arbitrary unitary operation.

A quantum gate is called *parametric* if its action depends on a continuous parameter. An example is the single-qubit rotation around the $Y$ axis:

$$R_Y(\theta) = \begin{bmatrix} \cos(\theta/2) & -\sin(\theta/2) \\ \sin(\theta/2) & \cos(\theta/2) \end{bmatrix}.$$

### B.2 AMPLITUDE ENCODING

Given a normalized real-valued vector $\mathbf{x} \in \mathbb{R}^{2^n}$ such that $\|\mathbf{x}\|_2 = 1$, *amplitude encoding* maps it into a quantum state as follows:

$$|\phi(\mathbf{x})\rangle = \sum_{i=0}^{2^n-1} x_i|i\rangle.$$

This encoding technique allows for representing $2^n$ classical values using only $n$ qubits, providing an exponential compression of input data, but requires $\mathcal{O}(2^n)$ gates in the general case Zhang et al. (2022).

## C  QNeRF Scalability: Parameters and Amplitudes

We analyse the scalability of the proposed Full QNeRF and Dual-Branch QNeRF models by examining how their complexity increases with the number of qubits. Specifically, we compare the number of amplitudes, classical parameters, and quantum gates required by each architecture for increasing values of $n$. A summary of these results is presented in Table 1.

As the number of qubits increases, the complexity of the Full QNeRF model grows exponentially not only in terms of the number of output amplitudes ($2^n$), but also in the number of classical parameters. This is because the encoding MLP must produce an output vector of size $2^n$, one amplitude per computational basis state. In our experimental setup, which uses a 3-layer MLP with a hidden dimension of 256, using up to 10 qubits results in fewer total parameters than those found in a standard classical NeRF model.

In contrast, the Dual-Branch QNeRF architecture exhibits a more favourable scaling pattern: it only requires $2^{n/2}$ amplitudes per branch. Although it uses two encoding MLPs—leading to a larger parameter count for small $n$—this approach allows the model to scale up to 18 qubits while maintaining a total parameter count comparable to that of classical NeRF. Importantly, the number of amplitudes remains within the feasibility range for implementation on near-term quantum devices. A critical factor in practical scalability is the preparation of the initial quantum state via amplitude encoding. In the general case, the number of quantum gates required for this preparation scales linearly with the number of amplitudes (Zhang et al., 2022). Under this assumption, the Dual-Branch QNeRF can be feasibly implemented on NISQ devices with up to 12 qubits per branch.

Recent work (Pagni et al., 2025) suggests that under certain structural assumptions, the cost of amplitude state preparation can be significantly reduced. An additional promising direction is Approximate Amplitude Embedding (Nakaji et al., 2022), in which a parameterised quantum circuit is trained to approximate the desired initial state. This approach has already shown promising results in a quantum computer vision task (Gharibyan et al., 2025) and may also benefit quantum NeRF architectures.

Another relevant factor is the number of variational parameters $\theta$ in the quantum ansatz, as this directly influences the number of circuit evaluations required to estimate gradients during training, typically via the parameter-shift rule (Wierichs et al., 2022). In our ansatz, each gate is associated with a single trainable parameter, making the total number of gates equal to the number of variational parameters. This number scales quadratically with the number of qubits, but in our setup it remains below 100, making the circuit suitable for current quantum hardware. Nonetheless, the adoption of gradient-free optimisation methods such as COBYLA (Powell, 1994) or SPSA (Spall, 1992) could further reduce the shot complexity during training.

## D  Training Details

We trained all models using Adam optimiser (Kingma & Ba, 2014) with an initial learning rate of $5 \times 10^{-4}$ for all parameters but the ones for the output scaling layer, to which were assigned a higher learning rate of $0.01$. The learning rate is decreased with a multi-step scheduler up to $6.25 \times 10^{-5}$ ($1.25 \times 10^{-4}$ for the scaling layer). We used a batch size of 64.

Quantum circuits were simulated on CPU using the *PennyLane* framework (Bergholm et al., 2018) (version 0.37), employing the noiseless *default.qubit* backend. Each model was trained for up to 50 epochs and tested every 5. The training is stopped if the testing PSNR starts decreasing. As can be observed in Table 3a and 3b, all models reached convergence, apart from Full QNeRF on *Lego* and *Drums* scenes.

In Tables 4 and 5, we report the CPU time required for each model to reach convergence. GPU-based quantum simulation was not feasible due to current limitations in PennyLane's amplitude embedding. Model components were implemented in *PyTorch* (2.6) (Paszke et al., 2019). We expect that the integration with GPU-accelerated quantum simulators (e.g. (Stein et al., 2024; Schieffer et al., 2025)) could significantly reduce training time, potentially making simulated QML models more efficient than classical counterparts.

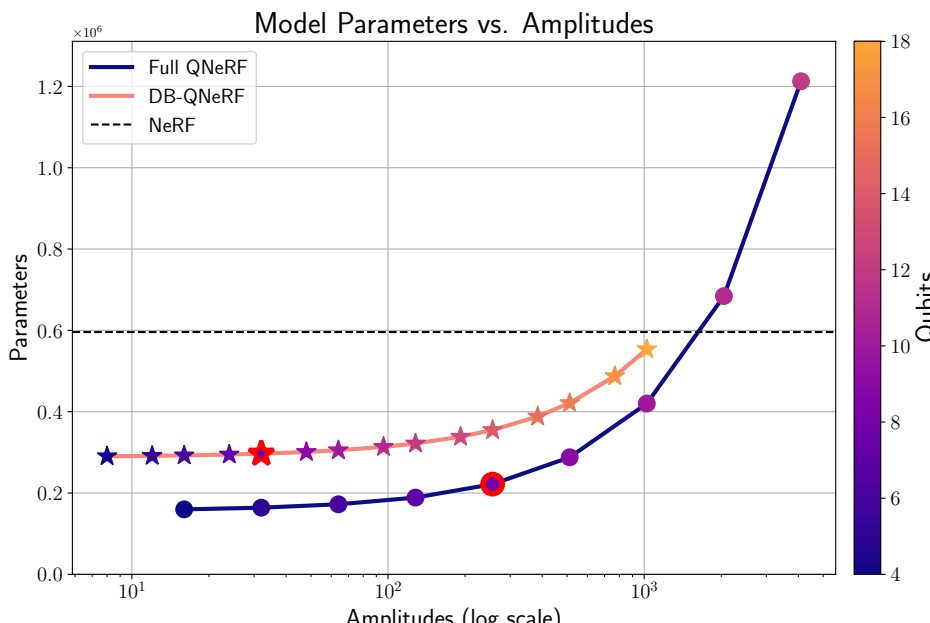

Figure 6: Comparison of Parameters vs Amplitudes for QNeRF models, where the Full is represented as a circle, and the Dual Branch as a star. The points highlighted in red correspond to the values for 8 qubits, as used in the experimental section.

| Model | Materials | Ficus | Lego | Drums |
|---|---|---|---|---|
| Full QNeRF | 40 | 43 | 50+ | 50+ |
| Dual-Branch QNeRF | 34 | 24 | 36 | 18 |
| Class. Baseline | 16 | 19 | 24 | 19 |

(a) Average stopping epoch across synthetic scenes.

| Model | Trex | Room | Horns | Fern |
|---|---|---|---|---|
| Full QNeRF | 28 | 39 | 42 | 42 |
| Dual-Branch QNeRF | 23 | 42 | 43 | 20 |
| Classical Baseline | 18 | 35 | 30 | 20 |

(b) Average stopping epoch across LLFF scenes.

Table 3: Summary of average stopping epochs for different models and datasets.

It is interesting to note that the time required by the Dual-Branch QNeRF is approximately $40\%$ more than the Full QNeRF model. This depends mostly on *PennyLane* framework, as the branched amplitude embedding is not currently supported (in our work, we prepared the initial state with two subsequent partial amplitude embeddings).

## E    ABLATION STUDY: "CLASSICAL" QNERF

As a simple ablation study, we want to evaluate if we increase in the performance of the proposed Full QNeRF model depends on the MLP feature encoding strategy, or on the usage of the QNN. To do so, we replace the QNN from the model described in Figure 2*a* with an MLP with 3 layers and a hidden dimension equal to 256, similar to the dimensions used in the MLPs in the quantum models, and in classical NeRF.

| Model | Materials | Ficus | Lego | Drums |
|-------|-----------|-------|------|-------|
| Full QNeRF | 39.36 | 42.31 | 49.20 | 49.20 |
| Dual-Branch QNeRF | 48.96 | 34.56 | 51.84 | 25.92 |

Table 4: Average training time (hours) to reach convergence on each synthetic scene.

| Model | Trex | Room | Horns | Fern |
|-------|------|------|-------|------|
| Full QNeRF | 3.41 | 4.06 | 5.71 | 2.02 |
| Dual-Branch QNeRF | 4.09 | 5.54 | 8.60 | 1.20 |

Table 5: Average training time (hours) to reach convergence on LLFF datasets.

In particular, the first MLP to learn the encoding is unchanged, as it is the ReLU plus the amplitude normalisations. However, in contrast to what is done for the QNeRF model, where the $2^n$ amplitudes are encoded in $n$ qubits, we process the $2^n$ "classical amplitudes" with another MLP, resulting in a "classical QNeRF". More in detail, the architecture of the second MLP consists of layers with dimensions $(2^n \times 256)$, $(256 \times 256)$, and $(256 \times 4)$ to obtain the RGB and $\sigma$ values at the end.

| Model | Materials | Ficus | Lego | Drums | Average |
|-------|-----------|-------|------|-------|---------|
| Full QNeRF | **33.88 ± 0.16** | **30.26 ± 0.21** | **34.47 ± 0.04** | **28.07 ± 0.05** | **31.67 ± 0.11** |
| DB QNeRF | 29.94 ± 0.31 | 28.59 ± 0.27 | 31.32 ± 0.25 | 25.63 ± 0.19 | 28.87 ± 0.26 |
| Class. Baseline | 29.90 ± 0.19 | 29.74 ± 0.17 | 31.79 ± 0.16 | 26.70 ± 0.13 | 29.53 ± 0.16 |
| "Class. QNeRF" | 23.11 ± 0.42 | 22.60 ± 0.06 | 21.16 ± 0.06 | 19.03 ± 0.05 | 21.47 ± 1.58 |

Table 6: Average PSNR over 5 seeds on the blender dataset, with the addition of the "Classical QNeRF" model (last row).

As reported in Table 6, the QNeRF model without the quantum component performs poorly, resulting in an average PSNR which is 8 points lower than the standard NeRF model. It is important to note that the performance drop does not depend directly on the number of parameters: as observed in Table 7, the "Classical QNeRF" has fewer parameters than the standard NeRF model, but more than both Full QNeRF and Dual-Branch QNeRF.

This shows that the quantum neural network in the proposed architecture is fundamental for the good performance of the model.

## F ADDITIONAL METRICS AND RENDERS

In the main paper, we report quantitative results in terms of Peak Signal-to-Noise Ratio (PSNR), as it is the most informative metric for evaluating pixel-wise reconstruction quality in novel view synthesis tasks. PSNR is defined as

$$\text{PSNR}(x, \hat{x}) = 10 \cdot \log_{10} \left( \frac{L^2}{\text{MSE}(x, \hat{x})} \right),$$

where $L$ is the maximum possible pixel value of the image (e.g., $L = 1$ for normalised images), $x$ is the ground-truth image, $\hat{x}$ is the reconstructed image, and MSE denotes the mean squared error.

Another relevant metric is the Structural Similarity Index (SSIM), which better correlates with human perception of image quality. SSIM between two images $x$ and $\hat{x}$ is defined as

$$\text{SSIM}(x, \hat{x}) = \frac{(2\mu_x \mu_{\hat{x}} + C_1)(2\sigma_{x\hat{x}} + C_2)}{(\mu_x^2 + \mu_{\hat{x}}^2 + C_1)(\sigma_x^2 + \sigma_{\hat{x}}^2 + C_2)},$$

where $\mu_x$ and $\mu_{\hat{x}}$ denote the mean pixel intensities, $\sigma_x^2$ and $\sigma_{\hat{x}}^2$ the variances, and $\sigma_{x\hat{x}}$ the covariance between $x$ and $\hat{x}$; $C_1, C_2$ are small constants for numerical stability.

| Model | Parameters ($k$) |
|---|---|
| Full QNeRF | 222 |
| Dual-Branch QNeRF | 297 |
| Classical NeRF | 590 |
| "Classical QNeRF" | 352 |

Table 7: Model parameters for proposed quantum models, Classical NeRF, and "Classical QNeRF".

We report SSIM values for the Blender dataset in Table 8a, using the same experimental setting as for PSNR. We observe that the trends across models are consistent with those obtained with PSNR, confirming that our proposed approach achieves superior perceptual reconstruction quality.

In addition, we provide qualitative results in Figure 7, showing example renders from the Blender dataset. Pixels with higher reconstruction errors are highlighted in red, providing further insights into the regions where differences among the models are most pronounced.

| Model | *Materials* | *Ficus* | *Lego* | *Drums* | Average |
|---|---|---|---|---|---|
| Full QNeRF | **0.9834±0.001** | **0.967±0.001** | **0.984±0.000** | **0.955±0.001** | **0.972±0.001** |
| DB QNeRF | 0.964± 0.002 | 0.954±0.001 | 0.966±0.002 | 0.926 ±0.003 | 0.953±0.016 |
| Class. Baseline | 0.964±0.002 | 0.955±0.004 | 0.973±0.002 | 0.939±0.003 | 0.958±0.012 |

(a) SSIM on the Blender dataset, under the same experimental setting given in Section 5.

## G    NOISE RESILIENCE: EXPERIMENTAL SETUP AND EXTENDED ANALYSIS

We provide the detailed methodology and additional results underpinning the noise resilience analysis presented in Section 5. The purpose of these experiments is to assess how the different ansatz architectures respond to realistic hardware noise.

To emulate the effect of noise on real quantum hardware, we employed IBM's publicly available noise models `FakeKyiv` and `FakeTorino` on *qiskit*(Javadi-Abhari et al., 2024). These models incorporate gate-dependent depolarisation, readout errors, and device-specific connectivity constraints. By applying them at the simulation level, we are able to obtain reproducible fidelity estimates under consistent noise conditions. Fidelity is a standard measure of similarity between two quantum states. For two density matrices $\rho$ and $\sigma$, it is defined as:

$$F(\rho, \sigma) = \left( \text{Tr}\sqrt{\sqrt{\rho}\,\sigma\,\sqrt{\rho}} \right)^2,$$

where $F(\rho, \sigma) \in [0, 1]$, with $F = 1$ indicating identical states and $F = 0$ corresponding to orthogonal states.

In particular, for pure states,

$$\rho = |\psi_\rho\rangle\langle\psi_\rho| \quad \text{and} \quad \sigma = |\psi_\sigma\rangle\langle\psi_\sigma|,$$

the fidelity reduces to

$$F(\rho, \sigma) = |\langle\psi_\rho|\psi_\sigma\rangle|^2.$$

This metric allows us to quantitatively assess how closely the noisy quantum state $\rho_{\text{noisy}}$ approximates the ideal noiseless state $\rho_{\text{ideal}}$ after circuit execution.

In our experiments, all amplitude embeddings employ Qiskit's `initialize` routine, followed by decomposition into the device's native gate set with four rounds of operator expansion (`decompose(reps=4)`) to decompose the gates into the set of gates that are feasible to the hardware under consideration. This ensures that the reported fidelities account for the practical cost of state preparation.

Also, other gates must be decomposed into gates from the base set of gates of the hardware. In particular:

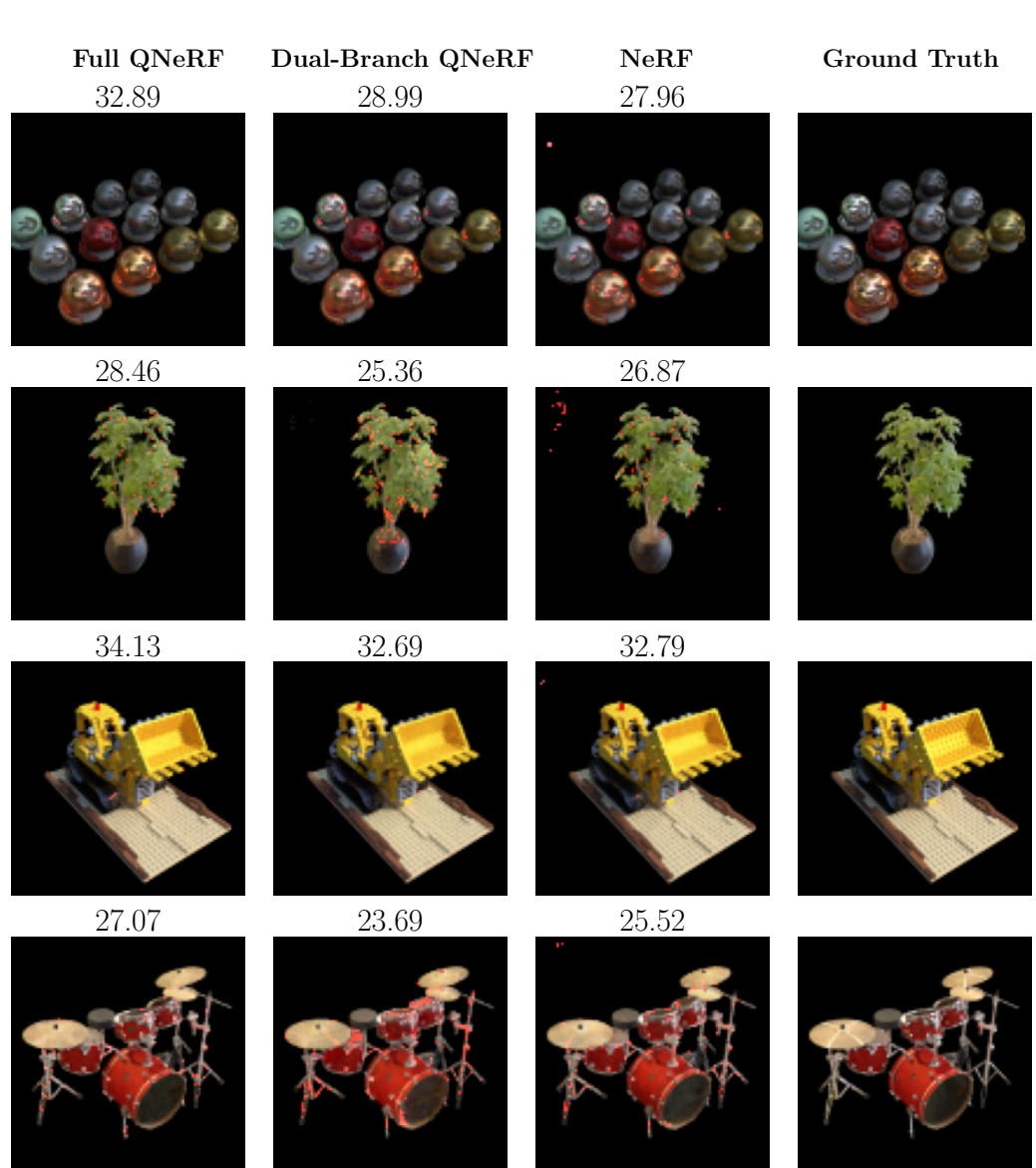

Figure 7: Visualisation of scenes from the Blender dataset (PSNR with respect to GT on the top). Pixels with higher error values are highlighted in red.

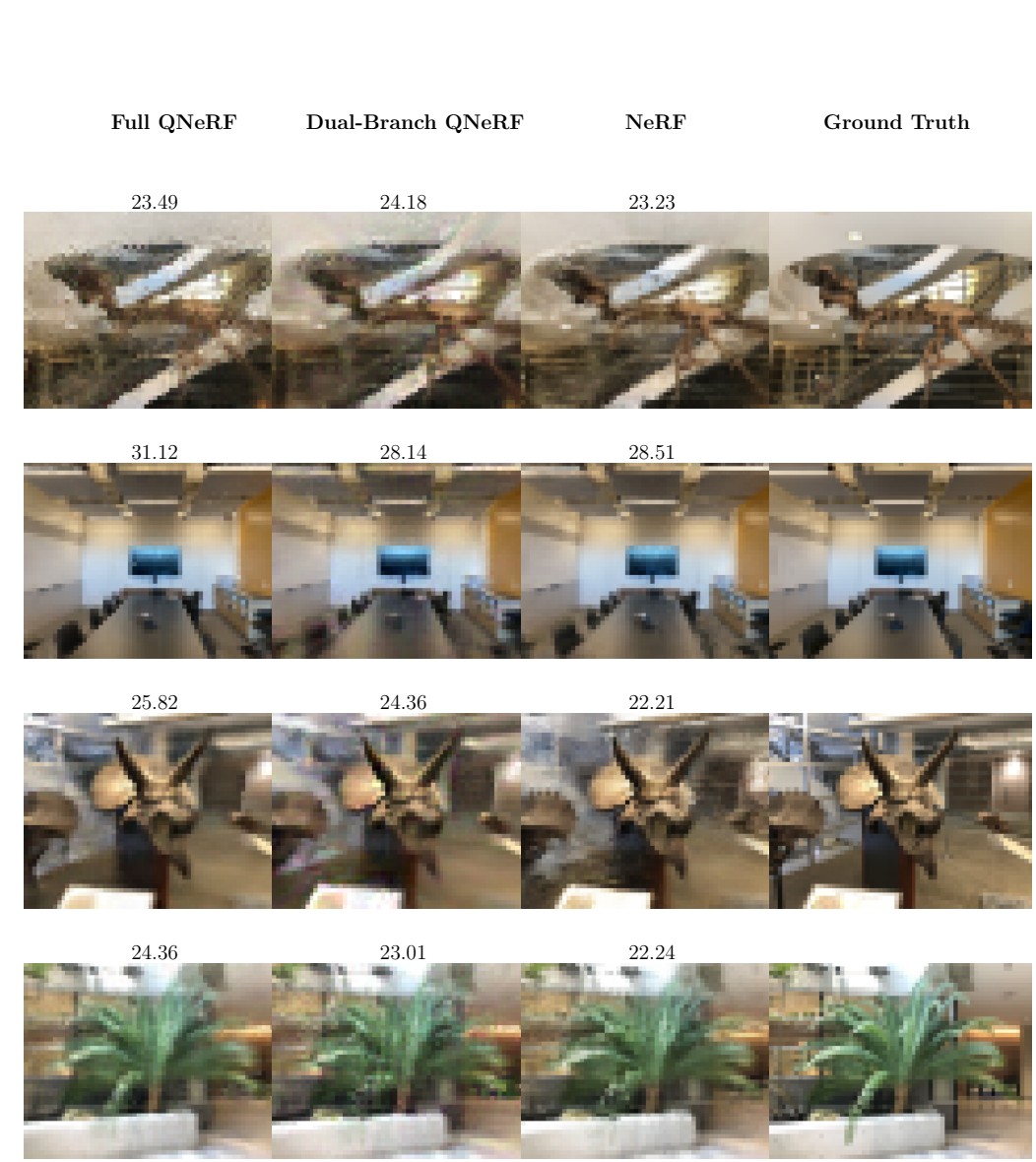

Figure 8: Visualisation of scenes from the LLFF dataset.

- *FakeKyiv* requires the circuit to be decomposed into sx, rz, ecr, and x gates, corresponding to $\sqrt{X}, R_z$, the echoed cross-resonance (ECR), and $X$ gates.

- *FakeTorino* requires gates sx, rz, cz, and x gates, corresponding to $\sqrt{X}, R_z$, controlled-$Z$, and $X$ gates.

For each configuration, we evaluated the circuit output under both noiseless and noisy conditions. The primary metric is the *state fidelity* between the noiseless state $\rho_{\text{ideal}}$ and the noisy state $\rho_{\text{noisy}}$ obtained after amplitude embedding, transpilation, and execution under the noise model. To mitigate bias from specific parameter values, we repeated the experiment 50 times with randomly drawn parameter initialisations and report aggregated fidelity statistics in Fig. 5.

The key finding is that the Dual-Branch ansatz achieves systematically higher fidelities compared to the Full-Branch model at equal depth. This advantage originates from its reduced reliance on exponentially large amplitude vectors and its more localised embedding strategy, which leads to a lower effective error rate.

The baseline case $\ell = 0$, corresponding to the fidelity of the encoding, is highlighted in Fig. 5. For the Full model, this corresponds to amplitude embedding on all qubits, while for the Dual-Branch model, it corresponds to separate embeddings on each half of the register. The intermediate case $\ell = 1$ in the Dual-Branch configuration is reported for completeness but is not representative of the architecture adopted in this work, as it lacks the full view-dependent interaction layer.

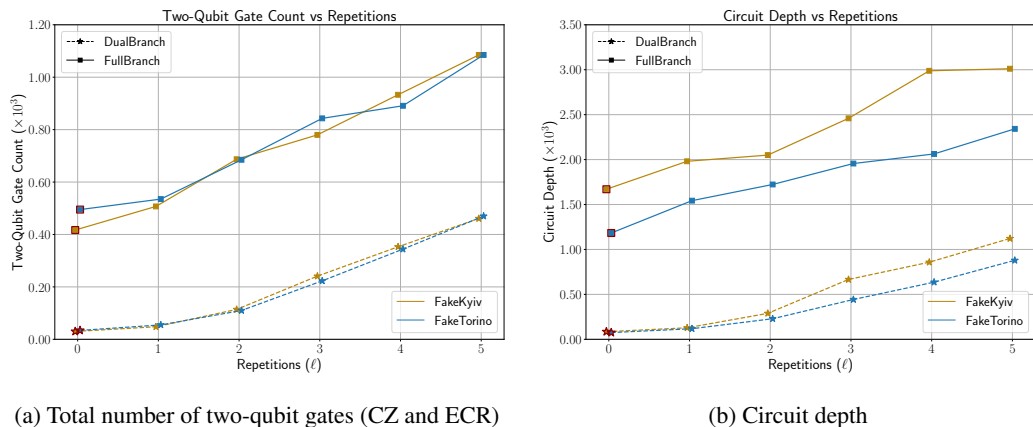

(a) Total number of two-qubit gates (CZ and ECR)  (b) Circuit depth

Figure 9: Evaluation of the two-qubit gates and circuit depth for different values of $\ell$, for Full and Dual-Branch models, and for different hardware. The numbers of repetitions selected in this work are $\ell = 1$ for the Full model and $\ell = 2$ for the Dual Branch.

To better understand the source of noise sensitivity, we extracted gate statistics from the transpiled circuits (Figure 9).

The analysis confirms that Dual-Branch circuits systematically require fewer two-qubit gates than Full circuits of comparable $\ell$ (Figure 9a). Since two-qubit gates are the dominant source of error in superconducting architectures, this reduction directly translates into higher state fidelities. Also, the gate depth is less than one third then the one required by the Full model for the values of $\ell$ considered in this work (Figure 9b).

It is important to stress that the reported fidelities are conservative estimates:

- No optimised transpilation strategies beyond the default Qiskit pass manager (with $optimization\_level = 4$) were employed (Murali et al., 2019; Li et al., 2019).

- No circuit-level error mitigation (e.g., Pauli twirling (Wallman & Emerson, 2016) Clifford twirling (Tsubouchi et al., 2025)) was applied.

- No post-processing techniques such as readout error mitigation or HAMMER (Tannu et al., 2022) were considered.

In realistic hardware runs, these methods are routinely applied and are known to significantly improve fidelities. Moreover, when training is performed directly on noisy hardware, model parameters may adapt to partially compensate for systematic error patterns.

## H  VARYING NUMBER OF QUBITS

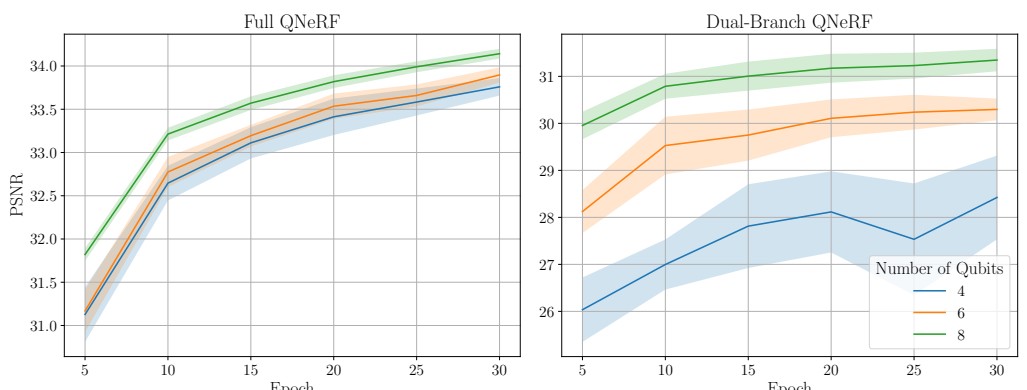

Figure 10: Training curves (testing PSNR over epoch) for the *Lego* scene for different number of qubits. The shadowed area represents the average $\pm$ the standard deviation.

We evaluate how the number of qubits $n$ impacts the performance of our method on the *Lego* scene (see Figure 10). While the Full QNeRF model shows competitive performance for $n = 4$ and $n = 6$, the Dual Branch model requires 8 qubits to perform comparably to the classical counterpart. Notice that DB QNeRF with $n = 6$ has the same number of amplitudes encoded as the Full model with $n = 4$. The latter has more parameters but a lower performance, probably due to the lower expressivity of the model (e.g., due to the different entangling layer structure). On the other hand, it is interesting to note that the classical parameter counts in DB QNeRF for $n = 4, 6, 8$ are $290k$, $293k$, and $297k$, respectively. This suggests that the dramatic increase in the performance depends predominantly on the increase in the representation capacity of the quantum component.

Finally, we visualise views for the tested models in Figure 11. Novel-view renderings by Full QNeRF show close to no perceptual difference in the reconstruction quality for different $n$.

## I  ESTIMATED EXECUTION TIME ON *IBM_Torino*

Starting from the analysis in App. G, we provide the expected execution time on the *IBM_Torino* gate-based quantum device (IBM Quantum, 2025). In practice, the transpilation process for a specific hardware produces the so-called Instruction Set Architecture (ISA) circuit representation, where each gate has a specific time duration. In this way, we are able to estimate the duration of a circuit execution on a target hardware.

In practice, the execution time provides a lower bound on the time that would be required to execute our proposed models on this specific quantum hardware. In real-world applications, the total time would increase, as in hybrid algorithms, quantum and classical devices must exchange information. Also, the time on a given device depends on additional factors, such as compilation and transpilation pipelines. Moreover, note that different quantum hardware can have substantially different execution times.

Figure 12 (a) shows the average time required to execute the considered quantum circuits with $\ell = 1$ for the Full QNeRF, and $\ell = 2$ for the Dual-Branch model. The number of qubits $n$ varies from 4 to 10, to assess scalability. Note that for the Full QNeRF model, most of the execution time depends on the state preparation. In particular, by using exact state preparation (i.e. the amplitude embedding routine from *qiskit*, we observe an exponential increase in the total time. This is expected,

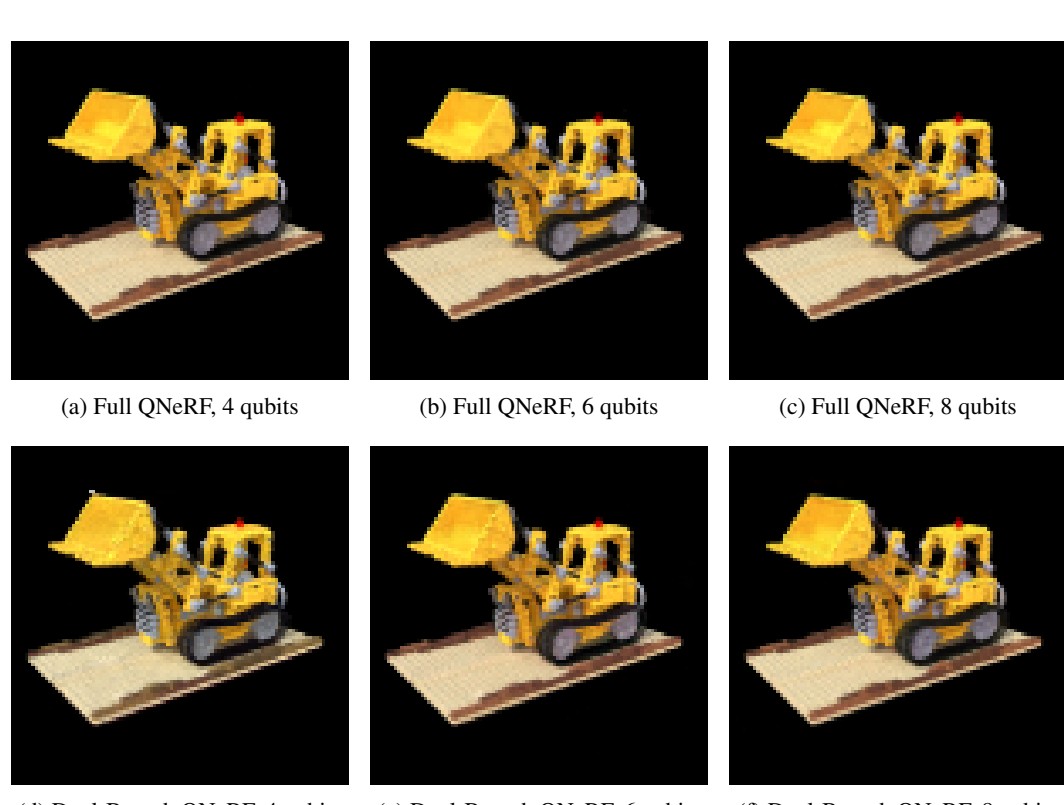

(a) Full QNeRF, 4 qubits      (b) Full QNeRF, 6 qubits      (c) Full QNeRF, 8 qubits

(d) Dual-Branch QNeRF, 4 qubits    (e) Dual-Branch QNeRF, 6 qubits    (f) Dual-Branch QNeRF, 8 qubits

Figure 11: Rendered output for the ablation study across models and qubit counts. The first row shows Full QNeRF results for 4, 6, and 8 qubits. The second row shows the corresponding Dual-Branch QNeRF reconstructions.

as amplitude embedding requires, in general, up to $2^n$ base operations. On the other hand, DB QNeRF requires much less time, as the partial amplitude embedding requires exponentially fewer operations than the standard amplitude embedding.

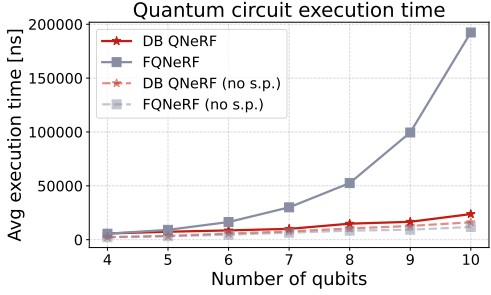
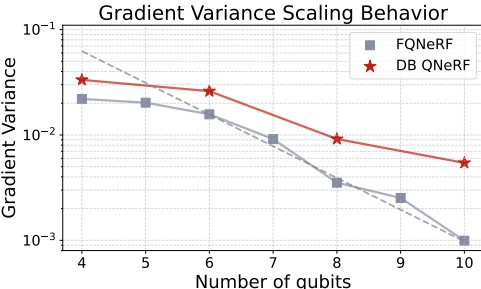

(a) Average execution time for 100 circuits with qubit counts from 4 to 10. Dashed lines indicate post–state-preparation time.

(b) Average gradient variance of the proposed models for Blender scenes. The dashed line references the value $2^{-n}$.

Figure 12: Comparison of gradient variance and quantum execution time for the evaluated models. Panel (a) presents expected execution times on *IBM_Torino* for circuits of increasing size, while panel (b) reports the variance of gradients on simulated hardware across scenes from the Blender dataset.

In practice, in our use case for $n = 8$, the DB model requires $3.7\times$ less time than Full ($1.48 \times 10^4$ns vs $5.25 \times 10^4$ns) for one execution.

## J    GRADIENT ESTIMATION

We now evaluate empirically the initial gradient variance of our proposed models. A well-known limitation on QML model scaling is the so-called Barren Plateau (BP) (McClean et al., 2018), i.e. an effect during a QNN training when the variance of the gradient decreases exponentially with the size of the system BPs are believed to increase the number of shots required to estimate the gradient and, therefore, also the total cost of the algorithm with the number of qubits of the system.

BP is correlated with the expressivity of the ansatz: In general, more expressive models are more prone to exhibit lower gradient variance. In Figure 12-(b), we evaluate the average gradient variance over 100 random initialisations. As BP depends on the task under consideration (and the associated energy optimisation landscape), for each initialisation, we select 100 batches of size 64 from a task in the Blender dataset and evaluate the average of the gradient variance. By doing so, we estimate the gradient in the same setting as the one for the main experiments. We fix the parameter $\ell = 1$ for Full, and $\ell = 2$ for Dual-Branch, and select the number of qubits from the set $n \in \{4, 5, 6, 7, 8, 9, 10\}$.

The Full QNeRF architecture exhibits a behaviour implying BPs, i.e. an exponential decay of the variance. This is consistent with our observations on the limited scalability of this model. However, BPs can become a limitation for higher $n$: Even if the gradient estimation requires an exponential number of shots, in our experimental setting $2^n = 256$, which is still feasible. Many quantum hardware producers use pipelines optimised for a higher number of shots (e.g. the default shot number of *IBM_Torino* is 5000, which is more than enough to provide a faithful estimation of the gradient with 8 qubits). On the other hand, the DB architecture, due to its lower expressivity, exhibits better scaling properties. For $n = 8$, the gradient variance is approximately three times higher than for the Full counterpart ($9.28 \times 10^{-3}$ vs $3.41 \times 10^{-3}$), and approximately four times for $n = 10$ ($4.13 \times 10^{-4}$ vs $1.12 \times 10^{-4}$). Note that in our experiments on the DB model, we always assume that the number of qubits to encode positional and view-dependent features is the same. For this reason, we only consider even values of $n$.

## K    INFERENCE UNDER SIMPLE NOISE MODELS

We provide an evaluation of Full QNeRF under two ideal noise models. First, following the procedure in (Wang et al., 2025), we assume that circuit noise can be modelled as zero-mean Gaussian

perturbations with varying standard deviations in the parameters of parametric gates, where higher values correspond to higher hardware noise. In particular, we evaluate the effect of Gaussian noise on the inference capabilities on the *lego* scene for standard deviation $\sigma \in \{0.01, 0.05, 0.1\}$. We report in Table 9 both PSNR and SSIM over a seed of the *lego* dataset, and show some qualitative results in Figure 13 (top). Note that $\sigma = 0.01$ produces a slight degradation of the results, with minimal change in SSIM. On the other hand, $\sigma = 0.05$ produces a degradation of 2 dB in PSNR, with some visible visual artefact (Fig. 13, centre image).

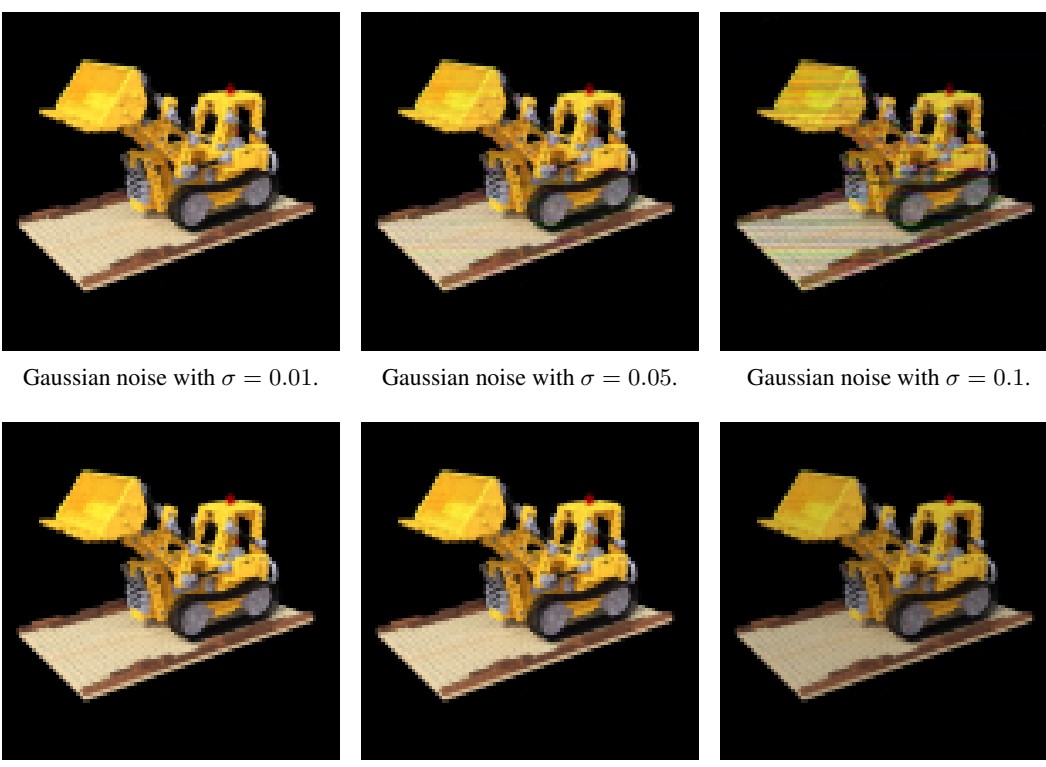

| Gaussian noise with $\sigma = 0.01$. | Gaussian noise with $\sigma = 0.05$. | Gaussian noise with $\sigma = 0.1$. |
| Readout error with $p = 0.001$. | Readout error with $p = 0.01$. | Readout error with $p = 0.1$. |

Figure 13: Novel views of the Full QNeRF model trained on the *lego* scene, under Gaussian perturbations (top row) and symmetric readout errors (bottom row).

Table 9: PSNR and SSIM values for different levels of Gaussian noise in Full QNeRF on *lego* scene.

| Gaussian noise | No noise ($\sigma = 0$) | $\sigma = 0.01$ | $\sigma = 0.05$ | $\sigma = 0.1$ |
|---|---|---|---|---|
| PSNR | 33.74 | 33.67 | 31.38 | 25.33 |
| SSIM | 0.9848 | 0.9847 | 0.9798 | 0.9477 |

Table 10: PSNR and SSIM values for different levels of readout error in Full QNeRF on *lego* scene.

| Readout noise | No noise ($p = 0$) | $p = 0.001$ | $p = 0.01$ | $p = 0.1$ |
|---|---|---|---|---|
| PSNR | 33.74 | 33.74 | 33.34 | 23.96 |
| SSIM | 0.9848 | 0.9848 | 0.9844 | 0.9556 |

Similarly, in Table 10 we evaluate the impact of readout error (i.e., bit flip) under the same assumptions. We provide some qualitative results in Figure 13 (bottom). In particular, we assume a probability of symmetric readout error for $p \in \{0.001, 0.01, 0.1\}$. We observe that $p = 0.001$ causes no degradation in the model performance. The value $p = 0.01$, comparable with current hardware readout errors, shows a small decrease in the performance. Finally, for $p = 0.1$ we observe a substantial performance

loss (i.e. the view appears darker, Fig. 13 bottom on the right). This depends on the fact that each output is shifted closer to zero (i.e., readout error maps each output channel $o_i \mapsto (1 - 2p)o_i$). For this reason, we suppose output scaling can mitigate readout error during training.

## L  MESH EXTRACTION

Similar to classical NeRF, QNeRF can be used to extract meshes from the trained models. As an example, we provide renders of extracted 3D meshes in Figure 14. Each image is a render of the 3D object extracted from resulting from a trained Full QNeRF model with 8 qubits using the Marching Cubes (MC) algorithm (Lorensen & Cline, 1987). The MC algorithm is implemented in the Python library *skimage*. The visualised objects are obtained with a resolution of $128^3$, and the sigma threshold of 10. Finally, the final rendering is obtained on the MeshLab software (Cignoni et al., 2008).

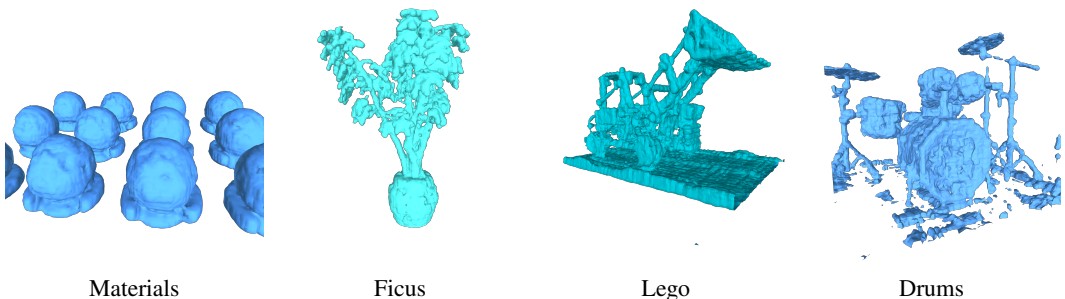

| Materials | Ficus | Lego | Drums |

Figure 14: MeshLab rendering of the 3D meshes extracted using Marching Cubes from the trained Full QNeRF model.

## M  ENHANCING QNeRF WITH CONICAL FRUSTUM AND IPE

NeRF established a new benchmark for novel-view synthesis. After its introduction, a wide range of extensions subsequently emerged, each targeting improvements in rendering efficiency, generalization, or robustness under challenging imaging conditions—for example, Mip-NeRF (Barron et al., 2021) for multi-scale rendering.

Although these more recent approaches vary in architectural detail, many retain the same foundational structure: one or more multilayer perceptrons are trained to approximate a volumetric radiance field from posed 2D observations. Subsequent works primarily differ in how they parametrize and sample the underlying volume. Among these, Mip-NeRF (Barron et al., 2021) replaces ray-based point sampling with a multi-scale, cone-based formulation that models the integration of signals across spatial frequencies when compared to original NeRF, yielding improved anti-aliasing and higher fidelity under varying resolutions.

This paper shows that augmenting NeRF with quantum neural networks can improve novel-view synthesis quality while simultaneously reducing the model size. Because the quantum components operate at the level of the underlying continuous signal representation (i.e., the underlying MLP), the resulting hybrid architecture remains compatible with a wide class of NeRF variants. This further motivates an investigation of whether the same quantum enhancements can be extended to more advanced formulations.

To this end, we examine the integration of QNN modules into Mip-NeRF (Barron et al., 2021). Compared to the original NeRF design, Mip-NeRF introduces two central modifications: (i) it replaces infinitesimal point samples with conical frustums, assigning each sample a finite footprint along the viewing ray; and (ii) it integrates the radiance field over these regions using a multiscale representation derived from integrated positional encoding (IPE).

We evaluate both classical mip-NeRF and a "Quantum-Mip-NeRF"' (QMip-NeRF) on the same setting used for the main experiment (single-scale Blender dataset downscaled to $100 \times 100$ pixels). The code for conical frustum sampling and IPE was adapted from (Mildenhall et al., 2022b). Each

other component (model structure, etc.) was not modified. We train each model for up to 15 epochs (requiring approximately 15 hours per scene), and we repeated each experiment 5 times. We report average PSNR for the scenes of the Blender dataset in Tab. 11.

Note that—despite not being trained on the multi-scale Blender dataset—the considered models can be used to represent scenes with higher resolution (i.e., by using a different scale). In Fig. 15, we provide visualisations on the *materials* scene to highlight differences in high-frequency details.

|  | *Materials* | *Ficus* | *Lego* | *Drums* | Average |
|---|---|---|---|---|---|
| QMip-NeRF | $32.43 \pm 0.25$ | $29.59 \pm 0.12$ | $33.25 \pm 0.11$ | $27.58 \pm 0.11$ | $30.71 \pm 0.15$ |
| Mip-NeRF | $30.53 \pm 0.69$ | $29.81 \pm 0.34$ | $32.73 \pm 0.27$ | $27.09 \pm 0.16$ | $30.04 \pm 0.37$ |

Table 11: PSNR after 15 epochs for "QMip-NeRF" and classical Mip-NeRF.

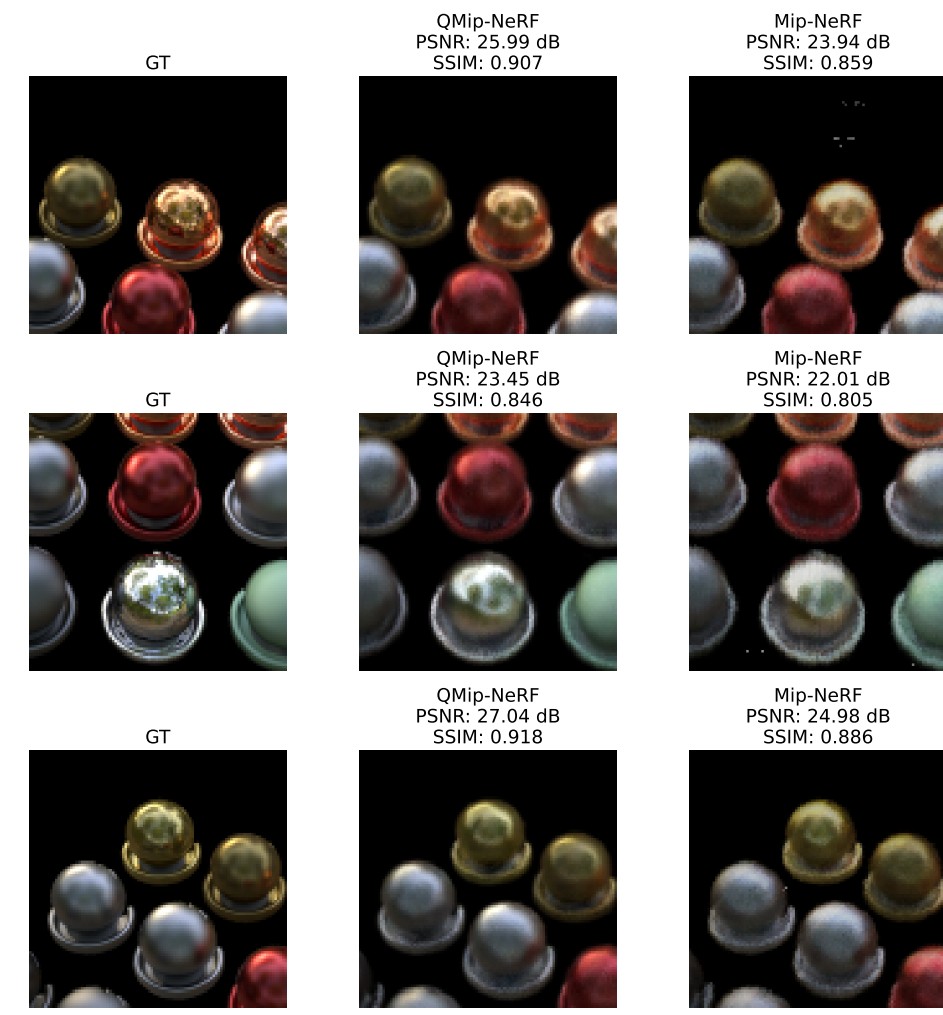

Figure 15: Visualisation of mip-NeRF variants obtained by inference on a higher resolution ($200 \times 200$ pixels) on the *materials* scene.

