# OpenReview forum: "Q-NeRF: Neural Radiance Fields on a Simulated Gate-based Quantum Computer"
_ICLR.cc/2026/Conference — Submitted to ICLR 2026_

### Official Review · Reviewer_wVBW · 2025-10-27

**Soundness:** 3
**Presentation:** 3
**Contribution:** 3
**Rating:** 6
**Confidence:** 1

**Summary:**

This work proposes the first hybrid quantum-classical model for novel-view synthesis of 2D images. It extends the Quantum Visual Fields (QVFs) approach to the domain of 3D scene representation, leveraging the superposition and entanglement properties of parameterized quantum circuits to achieve a more compact model representation than conventional NeRF.

**Strengths:**

1. Through amplitude embedding, Q-NeRF can represent $2^n$ classical values using only $n$ qubits, achieving exponential parameter compression.

2. The method overcomes the limitations of current noisy intermediate-scale quantum devices, particularly with the Dual Branch architecture, which significantly reduces state preparation complexity through branch-wise encoding.

3. Two architectures are provided, Full QNeRF and Dual Branch QNeRF, targeting maximal expressiveness and hardware compatibility, respectively.

4. Systematic experiments are conducted on the Blender and LLFF datasets, covering noise-free simulations, noisy environments, and scalability analyses. The experimental design is rigorous, including five repetitions with different random seeds to ensure the reliability of results.

**Weaknesses:**

1. Experiments are conducted only on downsampled images (100×100 pixels for the Blender dataset and 63×47 pixels for the LLFF dataset). It is unclear whether the method can be extended to high-quality, high-resolution scenes.
2. It remains uncertain whether existing approximate amplitude encoding schemes can truly overcome exponential complexity while maintaining performance. For higher-resolution scenes, such as 512×512 pixels, how many qubits would be required to achieve competitive results?
3. More advanced NeRF variants, such as Mip-NeRF and Instant-NGP, have already significantly outperformed the original NeRF in many aspects. Comparing only with the original version may not accurately reflect the competitiveness of Q-NeRF in the current technological landscape.
4. It is unclear whether the Q-NeRF approach can be extended to other 3D vision tasks. Tasks such as 3D reconstruction, scene editing, and dynamic scene modeling may also potentially benefit from quantum representations.

**Questions:**

See the Weaknesses.

---

> ### Author Response · Authors · 2025-11-21
>
> 1. In principle, there is nothing that prevents the application of our method to high-quality scenes. The reason why our experiments considered downscaled dataset is the simulation time (as higher resolution scenes have more samples and require longer training).
>
> 2. Note that as the model is learning a continuous quantum neural field, there is no straightforward connection between the model size and the scene resolution (in the sense that the scene is compressed in the ansatz unitaries). For instance, our QNeRF model with 8 qubits can be used to represent scenes with 512x512 pixel resolution or more. However, scenes with higher-frequency details at increasing resolution may require scaled-up models to obtain the same representation quality in novel views.
>
> 3. In this work, we explore the integration of a QNN in the NeRF architecture as a foundational challenge. Recent NeRF-based models are based on additional techniques that do not depend only on the underlying neural model. For example, mip-NeRF [1] has the same underlying principle of NeRF, but with a different sampling strategy, as it employs cones instead of rays. To show that our method is compatible with mip-NeRF and, possibly, other related methods, we are modifying our model to include other sampling strategies. We hope to show some results by the end  of the month within the reviewer-author discussion phase.
>
> 4. There is nothing that prevents the extension of QNeRF to other 3D tasks, as long as the underlying principle is based on Neural Radiance Fields. As an example, we performed mesh extraction with the marching cubes algorithm starting from the models trained for novel-view synthesis. We show the renders (from MeshLab) in the newly added Appendix L (see the revised paper draft). More complex tasks such as dynamic scene modeling could be investigated in future work (e.g., by introducing time dimension). They would  require scenario-specific design choices (e.g., inductive biases) that would justify a stand-along research project.
>
>
> [1] Barron et al. “Mip-NeRF: A Multiscale Representation for Anti-Aliasing Neural Radiance Fields.” ICCV, 2021.

---

> > ### Comment · Reviewer_wVBW · 2025-11-26
> >
> > Thank you for the authors’ response, which has resolved most of my concerns. I have also carefully reviewed the comments from the other reviewers. Since I am not an expert in this area, I maintain my original score.

---

### Official Review · Reviewer_ieL9 · 2025-10-28

**Soundness:** 2
**Presentation:** 3
**Contribution:** 2
**Rating:** 2
**Confidence:** 4

**Summary:**

The paper proposes QNeRF, a hybrid quantum–classical replacement of the NeRF MLP, with two variants (Full and Dual-Branch). Amplitude embeddings, local-Z readouts with parity averaging, and a learnable “de-concentration” scaling are used to mitigate trainability issues. On downscaled Blender/LLFF scenes, Full QNeRF achieves higher PSNR than a classical baseline with <50% parameters; Dual-Branch matches baseline PSNR while being more noise-tolerant under IBM FakeKyiv/FakeTorino noise.

**Strengths:**

Originality: The paper introduces QNeRF, a NeRF variant where the MLP is replaced by parameterized quantum circuits (PQC). Beyond the straightforward hybridization, the authors propose a Dual-Branch quantum embedding that separately encodes position and viewing direction before composition, explicitly targeting NISQ constraints (shallower circuits, reduced state-preparation burden). This is a creative combination of implicit neural representations and quantum feature maps that broadens how NeRF-like volumetric rendering can be parameterized.

Quality: On two datasets, QNeRF attains comparable or better PSNR than a classical NeRF baseline with fewer than half the learnable parameters, indicating a favorable accuracy–parameter trade-off. The results also show that the Dual-Branch variant preserves performance while aligning the circuit design with realistic hardware limits, which supports the claim of hardware-aware modeling.
Clarity: The paper is clearly written and easy to follow. The problem setup, the transition from classical MLPs to PQC blocks, and the end-to-end rendering pipeline are explained in a way that makes the contribution reproducible. Definitions and symbols are used consistently, and the motivation for the Dual-Branch construction is articulated with sufficient intuition.

Significance: QNeRF offers a new template for marrying NeRF-style continuous scene representations with quantum embeddings, potentially enabling future lines of work on quantum-accelerated volume rendering, quantum feature maps for 3D geometry, and hardware-aligned implicit modeling.

**Weaknesses:**

1. Limited practicality of the model. When the number of qubits is $n$, the overall complexity of QNeRF remains $2^n$, e.g., in the MLP replacement and state preparation. Even if the PQC step has complexity $\mathcal{O}(n)$, the end-to-end complexity is still $2^n$. Moreover, quantum computation typically requires many measurement shots and has inefficient backpropagation, so the wall-clock time of running QNeRF on a quantum computer may be much higher than classically simulating QNeRF.

   Suggestion: Optimize the model architecture so that the motivation for introducing quantum computation is more compelling. Alternatively, compute the end-to-end resource consumption of QNeRF and compare it against the time for classical simulation; specify the conditions under which QNeRF achieves quantum advantage, and then discuss the practical value of QNeRF.

2. Insufficient numerical experiments. Important dimensions are missing, such as how performance scales with the number of qubits, how results depend on the number of shots, and how noise affects inference/training.

   Suggestion: Add the above missing numerical studies to substantiate the claims.

**Questions:**

1. Scaling with qubit count and depth. How does QNeRF’s performance change with the number of qubits (n)? Please provide curves such as (n) vs. PSNR and layer depth vs. PSNR, and discuss the observed trends.
2. Conditions for practical quantum advantage. Under what parameter regimes—qubit count ($n$), layer depth ($l$), number of shots, noise strength, etc.—can QNeRF achieve practical quantum advantage? Please specify the conditions and provide supporting evidence or estimates.
3. Dual-branch claim on amplitude reduction. The paper states that, compared to the Full QNeRF approach, the dual-branch strategy reduces the number of amplitudes exponentially. However, the counts between the two schemes appear to have a quadratic relationship rather than an exponential one. Please clarify this claim and reconcile the discrepancy.
4. Noise resilience evaluation (Section 5.1). Section 5.1 evaluates state fidelity vs. layer depth $l$ only. To substantiate the noise-resilience claim, please add experiments that vary noise strength (e.g., readout/dephasing/coherent error rates) and measure their impact on inference/training performance (e.g., PSNR, gradients, convergence behavior).

---

> ### Author Response · Authors · 2025-11-21
> **Answers to Questions 1 and 2**
>
> 1. We evaluate the scaling w.r.t. qubit counts on the Lego scene, by selecting models with number of qubits $n=$ 4, 6 and 8. We report full PSNR curves per epoch for both Full and DB QNeRF in the newly added Appendix H. We observe that in the Full model, the increase in qubit counts results in a modest increase in performance (on average, less than 1 dB of PSNR at each stage of training). On the DB model, we observe a consistent performance increase with more qubits (e.g. the difference between 4 and 8 qubits after 5 epochs is of approximately 4 dB, going down to 3 dB after 30 epochs). The small difference in the parameter number (290k vs 297k) suggests the increase in performance depends on the increased number of amplitudes encoded, and in the representational power of the QNN. Also, in Appendix E, we consider the QNeRF model when the QNN is replaced by a classical MLP. The resulting “classical QNeRF” model exhibits a low performance and comparable parameter count (352k), suggesting that model performances depend highly on the QNN considered and not the classical amplitude embedding.
>
>
>
>
> 2. “Practical quantum advantage’’ remains a long-term objective for QML, and QNeRF is no exception. Our method ultimately would require large-scale quantum resources—that are not yet available—to potentially demonstrate practical quantum advantage. In practice, achievable performance depends critically on the underlying hardware architecture (connectivity, native gate set and noise in the system), the quality of compilation and routing, and the effectiveness of online and offline error-mitigation strategies.
>
> Under optimistic noise mitigation assumptions—assuming performance degradation on the order of a few dB—and fixing the qubit count to n = 8, we benchmarked the estimated execution on the IBM_Torino architecture. With default transpilation, a single execution of the Full QNeRF circuit requires approximately 5.2×10⁴ ns, whereas the Dual Branch variant requires 1.48×10⁴ ns (i.e., 3.5x faster).
> To illustrate the resulting resource demands, consider reconstructing a single view of a 100×100 image, with 194 rays per pixel and 1000 shots per circuit. Assuming access to 10 parallel copies of the circuit (i.e. by performing 10 parallel executions in a Quantum Multi-Programming fashion [1, 2]), feasible given IBM_Torino’s 133 qubits, the total hardware time would be:
> * Full QNeRF: ~2.8 hours
> * DB QNeRF: ~51.7 minutes
>
> From this point of view, we could achieve an “advantage” from the performance side, but impractical from computational time (consider that producing a view with a classical model has an inference time of less than a second). On the other hand, a device able to run 100 parallel executions (e.g. IBM’s planned 1000+-qubit systems projected for 2027), the time to produce a view would be ~16.8 min and ~5.2 min (assuming each execution takes care of a different subset of pixels, or a different view). So, in this regard, parallelization could drastically reduce computational requirements.
> A key bottleneck is the gate expansion introduced during compilation. The logical Full QNeRF circuit contains 36 gates (plus state preparation), yet its ISA-level representation requires nearly 600 physical gates (most of them required for state preparation). On the DB model, starting with 70 gates and the reduced state preparation, we obtain ISA representation of 100 physical gates (approximately 20 required for state preparation).
>
> This blow-up substantially impacts runtime of the quantum part. We see this mismatch between algorithmic circuits and hardware-native gate sets as the predominant obstacle to short-term practicality. Consequently, future work targeting near-term quantum advantage should prioritize hardware-aware circuit design, explicitly incorporating native operations and connectivity constraints [3]. Also note that this preliminary analysis is based on a specific quantum hardware of a specific quantum computational paradigm: different machines could exhibit substantially different properties (e.g., Quantinuum’s devices have an all-to-all connectivity [4], and could require much less quantum gates for the transpilation). In addition, approximate amplitude embedding [5] could also have a major role in reducing the total runtime.
>
> In conclusion: To assess practical advantage, it is also crucial to consider a metric (time or performance), and a target hardware and transpilation approach.

---

> ### Author Response · Authors · 2025-11-21
> **Answers to Questions 3 and 4**
>
> 3. We thank the reviewer for pointing out this lack of clarity. The number of amplitudes (and, asymptotically, of parameters and gates) grows as $a_{F}(n) = 2^n$ in the case of FQNeRF, and $a_{DB}(n) = 2^{n/2+1}$ for DB QNeRF. If we consider the “reduction” as $a_F(n)/a_{DB}(n) = 2^{n/2-1}$, which is exponential in n (i.e., we need “exponentially less amplitudes”). As correctly noticed by the reviewer, it also holds $a_F(n) \approx (a_D(n))^2$. We better reformulated section 4.1 to explain more carefully our statement (see the updated draft).
>
> 4. The fidelity estimation included in Section 5.1 provides some information of the noise resilience of our model under realistic noise models, i.e. fixed target hardware and full transpilation pipeline.
>
> In general, using the same noise model to evaluate a QML model of this size would be unfeasible in PennyLane, due the increase in resources required (for example, we would need to simulate thousands of gates instead of 70). For this reason, in similar works (e.g. [6]), the noise is simulated as random Gaussian perturbations with varying standard deviations added to gates’ parameters. We perform a small-scale evaluation of the impact of Gaussian perturbation with different standard deviations on the Full QNeRF model’s performance, evaluated as inference PSNR and SSIM, in the newly added Section K in the Appendix. We observe that Gaussian noise with $\sigma=0.01$ impacts slightly the model. Higher values such as $\sigma=0.05$ impact noticeably the performance, but lead to values comparable to the classical baseline. Finally, $\sigma=0.1$ results in a noticeable performance drop (8dB in PSNR).
>
> In addition to the Gaussian noise, we perform  another, more realistic evaluation of the readout error. This specific error does not depend (directly) on the transpilation, and is easy to simulate classically. We evaluate a symmetric readout error for $p=0.001, 0.01, 0.1$. Interestingly, probability of 0.01 (realistic within current hardware capabilities) provides almost no drop in performances. Moreover, as readout error with probability $p$ induces a shift of each expected output $o_i$  of the form $o_i\mapsto (1-2p)o_i$, we expect the final output scaling to partially mitigate this specific error, either during training or possibly as final finetuning process (e.g., the “noise-free” output can be obtained analytically by using as the scaling factor of the $i-$channel $\alpha_i\mapsto$ $\alpha_i/(1-2p)$).
>
>
>
>
>
> [1] P. Das, S. S. Tannu, P. J. Nair, and M. Qureshi, A case for multi-programming quantum computers, in Proceedings of the 52nd Annual IEEE/ACM International Symposium on Microarchitecture (2019) pp. 291–303
>
> [2] Baker, Jack S., et al. “Parallel Hybrid Quantum-Classical Machine Learning for Kernelized Time-Series Classification.” Quantum Machine Intelligence, vol. 6, no. 1, June 2024, p. 18. DOI.org (Crossref), https://doi.org/10.1007/s42484-024-00149-0.
>
> [3] Cheng, Jinglei & Wang, Hanrui & Liang, Zhiding & Shi, Yiyu & Han, Song & Qian, Xuehai. (2022). TopGen: Topology-Aware Bottom-Up Generator for Variational Quantum Circuits. 10.48550/arXiv.2210.08190.
>
> [4] Pino, J. M., et al. “Demonstration of the Trapped-Ion Quantum CCD Computer Architecture.” Nature, vol. 592, no. 7853, Apr. 2021, pp. 209–13. DOI.org (Crossref), https://doi.org/10.1038/s41586-021-03318-4.
>
> [5] Nakaji, Kouhei, et al. “Approximate Amplitude Encoding in Shallow Parameterized Quantum Circuits and Its Application to Financial Market Indicators.” Physical Review Research, vol. 4, no. 2, May 2022, p. 023136. DOI.org (Crossref), https://doi.org/10.1103/PhysRevResearch.4.023136.
>
>
> [6] Shuteng Wang, Christian Theobalt, and Vladislav Golyanik. Quantum visual fields with neural amplitude encoding. In Neural Information Processing Systems (NeurIPS), 2025.

---

### Official Review · Reviewer_vqrq · 2025-10-31

**Soundness:** 3
**Presentation:** 3
**Contribution:** 2
**Rating:** 6
**Confidence:** 3

**Summary:**

This paper proposes two novel models based on parameterized quantum circuits (PQCs) designed for learning 3D models, a task analogous to classical Neural Radiance Fields (NeRFs). The authors present a comprehensive performance comparison against classical methods, analyzing metrics like parameter count, gate count, and output fidelity (PSNR). Through numerical simulations, the paper demonstrates a significant performance advantage over classical benchmarks and includes an analysis of how noise impacts the models relative to circuit depth.

**Strengths:**

The paper proposes two novel quantum models that leverage parameterized quantum circuits for the complex task of learning 3D models.
The authors conduct a comprehensive comparative analysis of their proposed schemes, evaluating the number of parameters, gate count, and Peak Signal-to-Noise Ratio (PSNR).
Numerical simulations indicate that the proposed quantum models achieve a significant performance improvement compared to classical benchmark algorithms.
The study includes a relevant analysis of the models' performance under noise, investigating their relationship with the number of circuit layers.

**Weaknesses:**

1. The paper proposes two architectures, but the DB QNeRF model shows almost no discernible advantage in the presented results. In terms of critical metrics like accuracy and parameter count, it does not appear to justify its inclusion. The authors should either provide a stronger rationale for this second approach or focus the paper on the more promising model.

2. The claim of efficiency is based primarily on a reduced parameter count, which is an incomplete metric for the real-world cost of a quantum algorithm. The analysis should be expanded to include measurement shots and state preparation costs.

3. The paper does not consider to address the trainability of the proposed PQC architectures. Variational quantum algorithms are notoriously prone to barren plateaus, which makes training ineffective, especially as the system size scales. The authors should provide theoretical or empirical evidence to show that their model design can mitigate or avoid this issue, thereby ensuring its potential for scalability.

**Questions:**

See weakness

---

> ### Author Response · Authors · 2025-11-21
>
> 1. In the main paper, we show that DB QNeRF model achieves an accuracy comparable with the classical baseline, with a reduced parameter count, and with higher noise tolerance and scalability w.r.t. Full QNeRF architecture. In addition, in the App. G, we show that for our main experimental setting, DB QNeRF requires approximately 25% of 2-qubit gates compared to the Full model. To better show the reduced “cost” of DB QNeRF, we now provide additional comparisons:
> * App. I: (estimated) Execution time on IBM_Torino
> * App. J: Gradient estimation
>
> Both appendices (discussed in details below) show additional insights on the scalability of DB QNeRF.
>
> 2. We agree with the reviewer that parameter count is just one of possible metrics. In the newly added Appendix I, we estimate the time that would be required on the IBM_Torino device for executing a quantum circuit, according to the number of qubits and the model (Full or Dual-Branch). In addition, we compute how much amplitude embedding takes (i.e., state preparation). We observe that the DB model on 8 qubits requires 3.7x less time compared to the Full model, with increasing differences as the number of qubits increases (reaching an order of magnitude for 10 qubits). This shows that DB architecture scales better than the Full one.
>
> 3. To evaluate the trainability aspects of the proposed models, we evaluate if the gradient variance “shrinks” exponentially with the size of the system. In the newly added Appendix J, we observe that the Full model is prone to barren plateau phenomenon, which does not seem to be the same for the DB model. It is important to note that barren plateaus are a problem only when the size of the system increases: in particular, even if we require “exponentially many” samples to estimate the gradient, for the size of the system considered (i.e. n = 8) we would require a limited number of shots. Note that many quantum hardware providers have a default number of shots that is order of magnitudes higher (e.g. IBM has a default number of shots = 5000). On the other hand, the DB model suggested better trainability, due to higher variance of the gradient (lines 1428-1430 page 27, and also Fig. 12 (b))

---

### Author Response · Authors · 2025-11-21

We thank the reviewers for their comments that will help to improve our work. We provide a first update of the paper with some of the suggested analysis and requested experiments. The revised text in the main paper is highlighted in red. Also, we added Sections H (a small ablation study on the number of qubits), I (evaluation of the expected execution time on a gate-based quantum computer IBM_Torino), J (Numerical gradient estimation), K (inference under simple noise models) and L (renderings of the 3D meshes extracted from the trained QNeRF models) in the Appendix. Note that only the titles of these new sections are highlighted in red. In the following, we provide detailed responses to the raised concerns below each review.

---

### Author Response · Authors · 2025-12-01

We provide a final update to the manuscript to address whether our proposed method and architecture can be combined with classical techniques beyond basic NeRF. We answer this affirmatively by integrating conical frustum sampling (instead of ray sampling) and Integrated Positional Encoding into our model, following the approach introduced in Mip-NeRF. The results, presented in the new Appendix M, show that the resulting “Quantum-Enhanced Mip-NeRF” achieves improved performance compared to the classical Mip-NeRF baseline.
This further supports our claim that QNeRF can serve as a backbone for a variety of NeRF-based architectures, paving the way for quantum-enhanced models for volumetric scene-representation tasks, analogous to the role played by QVF [1] in 2D settings.

We will integrate the obtained results and the insights gained into the main paper upon acceptance.

[1] Shuteng Wang, Christian Theobalt, and Vladislav Golyanik. Quantum visual fields with neural amplitude encoding. In Neural Information Processing Systems (NeurIPS), 2025.

---

### Author Response · Authors · 2025-12-02
**Summary of Updates Added in the Revised Manuscript**

We thank the reviewers once again for their constructive feedback. In the revised manuscript, we have added six new appendices:
* H: Ablation study on the number of qubits/amplitudes
* I: Evaluation of execution time on real quantum hardware
 * J: Gradient-based analysis to assess potential trainability issues in the considered models
*  K: Evaluation of the reconstruction fidelity of the proposed FQNeRF model under two different noise models (Gaussian Perturbation and Readout Error)
 * L: Visualization of mesh extraction from the trained QNeRF model using the Marching Cubes algorithm
 * M: Integration of QNeRF with Mip-NeRF enhancements (conical frustum sampling and Integrated Positional Encoding)

Appendices I, J, and K address questions concerning scalability, practical usability, and noise resilience. They suggest that large-scale QML models will require quantum hardware with higher connectivity and/or improved transpilation methods before practical quantum advantage becomes achievable, while also indicating that model inference can yield competitive performance under reasonable noise assumptions.

Conversely, Appendices L and M demonstrate that the proposed QNeRF architecture can indeed serve as a backbone for various NeRF-based models. We show that it supports mesh extraction via established classical techniques and can be composed with more advanced NeRF architectures employing alternative sampling and encoding strategies, such as Mip-NeRF. Notably, QMip-NeRF achieves improved performance relative to the classical Mip-NeRF baseline on the dataset considered.

We hope these updates enhance the clarity and overall quality of the work.

---

### Meta-Review · Area_Chair_tv4c · 2026-01-06

**Summary:**

The paper proposes QNeRF which replaced the MLP in NeRFs with learned quantum circuits.
The combination of QuantumML and NeRF is novel, but the gap between theoretical novelty and practical utility make the submission premature for a top-tier venue. Even though QNeRF is parameter-efficient, the actual rendering time is impractical, taking several hours for a single low-res view that baselines finish in seconds.

**Reviewer Concerns:**

See above.

**Reviewer Scores:**

No changes expected. The low-confidence score of 6 is explicitly not changing, while the other reviews are unlikely to change.

---

### Decision · Program_Chairs · 2026-01-26

Reject